# TAH-Quant: Effective Activation Quantization in Pipeline Parallelism over Slow Network

## Abstract

Decentralized training of large language models offers the opportunity to leverage computational resources across geographically distributed participants, but is often bottlenecked by network communication, particularly in pipeline-parallel settings. While pipeline parallelism partitions model layers across devices to handle large-scale models, it requires frequent communication of intermediate activations, which can be challenging when network bandwidth is limited. To address these issues, we propose TAH-QUANT (**T**ile-wise **A**daptive **H**adamard **Quant**ization), a novel activation quantization framework for pipeline parallelism. TAH-QUANT integrates fine-grained tile-wise quantization, entropy-guided tile-wise adaptive bit allocation for bit usage, and a Hadamard-based transformation with pivot swapping to effectively suppress outliers. Under a bounded relative quantization-error condition (Assumption 4.4), which we verify empirically, we prove that pipeline parallel training equipped with TAH-QUANT maintains a convergence rate of $\mathcal{O}(1/\sqrt{T})$, matching that of vanilla stochastic gradient descent. Extensive experiments demonstrate that TAH-QUANT aggressively compresses activations to an effective 3–4-bit precision, providing up to $5.4\times$ throughput speedup over uncompressed FP32 and up to $1.33\times$ wall-clock speedup over AQ-SGD, while preserving training convergence, avoiding AQ-SGD's activation-cache overhead, and generalizing across the fine-tuning, instruction-tuning, and from-scratch pretraining settings we evaluate.

## 1 Introduction

Decentralized or open collaborative training of large language models (LLMs) has recently gained significant attention as it enables the pooling of computational resources across multiple geo-distributed participants, thus facilitating the training of models that exceed the capacity of any single resource contributor Ryabinin & Gusev (2020); Yuan et al. (2022); Gandhi et al. (2024). However, a major barrier to these approaches is network communication: unlike specialized clusters equipped with high-speed interconnects, decentralized settings typically rely on slower networks, severely constraining training efficiency Wang et al. (2022; 2023b). On the other hand, scaling LLM training in the state-of-the-art scale necessitates distributed parallel training — particularly pipeline parallelism Huang et al. (2019); Narayanan et al. (2019; 2021), which partitions model layers across multiple stages to support training large-scale models. Yet, pipeline parallelism inherently requires frequent transmission of activations and the corresponding gradients between adjacent pipeline stages. In this paper, we explore *how to effectively compress the communication volume to accommodate pipeline parallelism over a slow network.*

Enabling efficient activation compression for pipeline parallelism over slow network connections has significant implications for democratizing large-scale LLM training Yuan et al. (2022); Wang et al. (2022). Currently, the capability to train state-of-the-art models remains concentrated among institutions equipped with specialized high-performance computing resources. Effectively addressing network communication bottlenecks would substantially reduce barriers to participation, allowing a broader array of contributors, including universities, startups, and individuals, to collaboratively train or fine-tune LLMs Douillard et al. (2025).

On the other hand, a significant obstacle arises from the fact that naive activation compression (e.g., quantization Han et al. (2016); Hubara et al. (2017)) methods can negatively affect training convergence.

Unlike gradient compression in data parallelism — where quantization errors typically behave as unbiased noise compatible with optimization procedures, compressing intermediate activations directly influences the neural network's forward computation, consequently introducing bias into gradient estimates in the later backward propagation Evans & Aamodt (2021); Chakrabarti & Moseley (2019). Specifically, in pipeline parallel training, compression errors incurred during activation transmission propagate through nonlinear transformations and can distort gradient calculations during the backward pass Wang et al. (2022). Thus, aggressively reducing activation precision without careful management will result in performance degradation or even training divergence.

To restrict the error propagation introduced by activation quantization, prior efforts, such as AQ-SGD Wang et al. (2022), have attempted to address this issue by compressing the changes in activations between training epochs rather than the activations themselves, thereby providing theoretical convergence guarantees leveraging the help from the error compensation. Although effective in preserving model accuracy, AQ-SGD requires storing previous activations for the whole dataset to compute these changes, resulting in substantial memory overhead. Such an approach poses practical limitations, especially in resource-constrained environments for the large-volumes of training data where storage capacity and system complexity are critical considerations.

In this paper, we solve this problem with a new approach for effective activation quantization in pipeline parallelism. In particular, we make the following key contributions:

**Contribution 1.** We propose TAH-QUANT, an activation quantization approach to alleviate communication bottlenecks in pipeline-parallel training of LLMs. Specifically, our method includes: (**i**) a fine-grained, tile-wise group quantization technique for localized precision control, effectively limiting quantization error; (**ii**) an entropy-guided, tile-wise adaptive bit allocation method that dynamically assigns precision based on activation distribution characteristics at a granularity that strictly refines token-level allocation, further optimizing the compression efficiency; and (**iii**) a Hadamard-based outlier suppression transform enhanced by a pivot element swap, which effectively mitigates quantization errors arising from extreme activation values. Collectively, these carefully designed techniques enable efficient, accurate low-bit quantization of activations, substantially improving the practicality of bandwidth-limited pipeline-parallel LLM training.

**Contribution 2.** We analyze the effect of TAH-QUANT's activation compression on pipeline-parallel optimization under standard stochastic assumptions, together with an empirically validated condition on the resulting compression error. Concretely, we prove that pipeline parallel training equipped with TAH-QUANT converges at a rate of $\mathcal{O}(1/\sqrt{T})$, matching that of vanilla SGD.

**Contribution 3.** We conduct extensive experiments on various LLM training tasks, including fine-tuning (`GPT-2XL`), instruction tuning (`Qwen2.5-3B`), and from-scratch pretraining (`LLaMA-3.2-1B`). We show that TAH-QUANT can aggressively quantize activations to 3–4 bits without sacrificing convergence relative to the state-of-the-art, *i.e.*, `AQ-SGD`, while introducing no additional activation-cache storage overhead. These results demonstrate that TAH-QUANT is effective across the various training tasks we study.

**Scope.** We focus on bandwidth-limited pipeline-parallel training. Our experiments use up to 16 GPUs, ranging from a single multi-GPU server to an 8-instance cluster, and emulate slow-network and distributed conditions—bandwidth, latency, and jitter with Linux `tc`, and node heterogeneity by throttling GPU clocks. Deployment over real wide-area geo-distributed networks with unreliable participants is left to future work.

## 2 Preliminary and Related Work

**Decentralized training of LLM.** Decentralized training of LLMs has garnered significant attention as an interesting attempt to democratize access to large-scale LLM training development Ryabinin & Gusev (2020); Borzunov et al. (2022; 2023); Gandhi et al. (2024); Blagoev et al. (2025). Early efforts demonstrated the feasibility of collaborative training across geographically distributed participants with constrained resources under the scope of data parallelism Diskin et al. (2021); Borzunov et al. (2022), where various effective gradient compression methods have been explored Wang et al. (2023b). To further scale out the training computation, more advanced modes of parallel strategies have been integrated Yuan et al. (2022); Ryabinin et al. (2023); Lu et al. (2024); Strati et al. (2024), for example, Yuan et al. (2022) addressed the challenges of training foundation models in heterogeneous environments by introducing a scheduling algorithm that optimally

allocates computational tasks across decentralized GPUs; Ryabinin et al. Ryabinin et al. (2023) proposed SWARM Parallelism, where temporary randomized pipelines between nodes are adaptively rebalanced to handle dynamic efficient training of large Transformer models using preemptive instances with limited network bandwidth. Douillard et al. Douillard et al. (2025) improve DiLoCo with sequential synchronization, comm–compute overlap, and quantized exchange. Most recently, Ramasinghe et al. Ramasinghe et al. (2025) propose *Subspace Networks*, which reduce model-parallel activation traffic by constraining projection weights to a shared low-rank subspace. This direction is complementary to TAH-QUANT: Subspace Networks reduce communication through weight-parameterization constraints, whereas TAH-QUANT is a drop-in activation-compression module that leaves the model parameterization unchanged.

**Activation compression in training.** Activation compression techniques have been studied to reduce memory and computational overhead in model training Liu et al. (2021); Bersatti et al. (2020); Georgiadis (2019); Fu et al. (2020); Liu et al. (2022); Chen et al.; Bian et al. (2024). Concretely, inherent sparsity in activation has been studied to minimize storage and computation in neural networks. For example, Zhang et al. Zhang et al. (2024) investigate the natural occurrence of sparse activations in pretrained Transformers and dynamically alternate between sparse and dense training phases to enhance pretraining efficiency Rhu et al. (2018); Jiang et al. (2022); Zhang et al. (2024); Li et al.. On the other hand, quantization-based methods Evans et al. (2020); Liu et al. (2021); Wang et al. (2023a) reduce the precision of activations to lower bit-widths, thereby decreasing memory usage. For example, Han et al. Han et al. (2016) presented Deep Compression, combining pruning, trained quantization, and Huffman coding. Hubara et al. Hubara et al. (2017) explore training neural networks with low-precision weights and activations. Chakrabarti et al. Chakrabarti & Moseley (2019) propose backpropagation with approximate activations for memory-efficient training. Chen et al. Chen et al. (2021a) introduced ActNN, employing 2-bit activation compressed training.

**Quantization for LLM.** Quantization has emerged as a key technique for serving LLMs efficiently by reducing the precision of weights Lin et al. (2024b); Frantar et al. (2022), activations Xiao et al. (2023), and KV-cache Liu et al. (2024c) for the process of generative inference, parameter-efficient fine-tuning Dettmers et al. (2023), and large-scale pretraining You et al. (2024); Liu et al. (2024a). For example, AWQ Lin et al. (2024b) quantizes the LLM weights by identifying a small subset of "salient" weight channels and scales them up before quantization, thereby preserving accuracy even at 4-bit weight precision. KIVI Liu et al. (2024c) proposes a tuning-free 2-bit quantization of the KV cache (with per-channel asymmetric scaling), dramatically reducing memory and enabling longer context lengths with negligible impact on generation quality. QLoRA Dettmers et al. (2023) demonstrated that a 4-bit quantized base model can be *fine-tuned* via low-rank adapters to reach the same performance as full `FP16` fine-tuning. LLM-QAT Liu et al. (2024a) introduces a data-free QAT scheme that allows 4-bit quantization of weights, activations, and even the KV cache while preserving performance for training. One essential problem in such quantization methods is how to effectively resolve the issues of outliers in the quantization group Lin et al. (2024a); Hu et al. (2025); You et al. (2024). An especially simple yet effective transformation for outlier suppression is the *Hadamard transform* Theodoridis & Koutroumbas (2009). Formally, the $N \times N$ Hadamard matrix $\mathbf{H}_N \in \{\pm 1\}^{N \times N}$ is defined such that $\mathbf{H}_N \mathbf{H}_N^T = N \mathbf{I}_N$ (so $\frac{1}{\sqrt{n}} \mathbf{H}_N$ is an orthonormal matrix). Multiplying a vector with dimension $N$ by $\mathbf{H}_N$ will evenly redistribute the vector's components across $N$ dimensions. Recent quantization studies leverage the *Hadamard transform* to suppress outliers and reduce quantization error, mainly for tensor-wise weight quantization in generative inference scenarios, such as QuaRot Ashkboos et al. (2024) and SpinQuant Liu et al. (2024b). In contrast, we apply lightweight Hadamard transforms for *activation communication* in pipeline-parallel training at a finer tile-wise granularity.

## 3 Activation Quantization

Pipeline-parallel training Huang et al. (2019); Narayanan et al. (2019; 2021) requires communication of intermediate activations between devices. This communication can become a key bottleneck on slow interconnects. To alleviate this by quantization-based compression, we leverage three carefully-designed mechanisms *in order to reduce the quantization error*, including: (**i**) fine-grained tile-wise group quantization for localized precision control (Section 3.1); (**ii**) an entropy-guided tile-wise adaptive bit-width allocation (Section 3.2); and (**iii**) a Hadamard-transform-based outlier suppression with a pivot element swap (Section 3.3).

We also discuss how we integrate the proposed TAH-Quant quantization method in pipeline parallel training in Section 3.5. We enumerate the details below.

### 3.1 Fine-Grained Tile-Wise Group Quantization

First, we introduce a fine-grained, tile-wise group quantization scheme for localized precision control. Specifically, instead of quantizing the entire activation tensor with a single set of parameters, we partition it into small tiles and quantize each tile independently. For example, consider an activation tensor $\mathbf{a}$ of shape $B \times S \times C$, i.e., $\mathbf{a} \in \mathbb{R}^{B \times S \times C}$, where $B$, $S$, and $C$ denote the batch size, sequence length, and number of channels (i.e., model dimension), respectively. We partition this tensor along the channel dimension into multiple tiles by grouping contiguous channels within each token. Each such tile (i.e., quantization group) can be noted as $\mathbf{a}_{i,j,t} \in \mathbb{R}^G$, where $G$ is the quantization group size determined by $G = \frac{C}{N_t}$, $N_t$ is the number of partitions of all the channels, and $i = 1, \ldots, B$, $j = 1, \ldots, S$, $t = 1, \ldots, N_t$ are the indices for each tile-wise quantization group. Note that each tile will form a separate quantization group with its own scale and zero-point. This approach ensures that each group is quantized using an optimal dynamic range, improving low-bit accuracy. By confining quantization error to small groups, we avoid the coarse tensor-wise scale being dominated by a few extreme values.

### 3.2 Tile-Wise Adaptive Bit Allocation

The fine-grained grouping addresses local range variation; however, even within a single token the activation values may have non-uniform importance: some contiguous channel windows are smooth while others contain sharp peaks. We therefore introduce an entropy-based, tile-wise adaptive precision allocation strategy that dynamically adjusts the quantization bit width at the granularity of small channel-window allocation tiles within each token rather than entire tokens.

**Tile decomposition.** Let $A \in \mathbb{Z}_{>0}$ denote the allocation-tile size, with $A \leq C$ and $A$ chosen to align with the quantization tile boundaries of Section 3.1. Each token is split into $M = C/A$ non-overlapping channel-window allocation tiles. We reshape the activation tensor

$$\mathbf{a} \in \mathbb{R}^{B \times S \times C} \longmapsto \tilde{\mathbf{a}} \in \mathbb{R}^{B \times (SM) \times A},$$

so that each row $\tilde{\mathbf{a}}_{i,j'} \in \mathbb{R}^A$ represents one allocation tile, indexed by $j' = 1, \ldots, SM$, that combines a token index and a channel-window index. Setting $A = C$ (so $M = 1$) recovers the previously studied token-level allocation as a strict special case; choosing $A < C$ yields finer precision control. In our default configuration, $A = G$, so each allocation tile is also a quantization group.

**Per-tile entropy.** For every allocation tile $\tilde{\mathbf{a}}_{i,j'} \in \mathbb{R}^A$ we form its normalized magnitude distribution

$$p_k = \frac{|\tilde{a}_{i,j',k}|}{\|\tilde{\mathbf{a}}_{i,j'}\|_1 + \epsilon}, \quad k = 1, \ldots, A, \tag{1}$$

where $\epsilon$ is a small positive constant for numerical stability, and define

$$\mathcal{H}(\tilde{\mathbf{a}}_{i,j'}) = -\sum_{k=1}^{A} p_k \log(p_k + \varsigma), \tag{2}$$

where $\varsigma > 0$ avoids $\log 0$. A larger $\mathcal{H}$ indicates more uniformly distributed tile energy, while a smaller value indicates stronger channel-wise outliers.

**Top-$p\%$ tile selection.** We rank the $SM$ allocation tiles of every sample by their entropy and assign `INT4` (high precision) to the top-$p\%$ *highest-entropy* tiles and `INT3` (low precision) to the rest. The intuition is unchanged from the token-level argument: high-entropy tiles lack a single channel that the Hadamard transform of Section 3.3 can isolate, so they need extra precision; low-entropy tiles concentrate energy on a few channels and thus tolerate aggressive quantization *after* the outlier-suppression transform. The resulting 1-bit per-tile bit map is packed with the quantized payload as metadata.

**Why tiles rather than whole tokens.** Empirically we observe that, within a single token, the entropy of different channel-window tiles can differ by more than an order of magnitude — some tiles contain a strong outlier while others are perfectly smooth. Whole-token allocation forces both kinds of tiles to share the same bit budget, over-allocating precision to tiles that can tolerate lower precision while under-allocating tiles that require higher precision. Tile-wise allocation avoids this mismatch: a token may now be encoded as a heterogeneous mixture of `INT4` and `INT3` tiles. Since tile-wise allocation reduces to token-level allocation when $A = C$ (every allocation tile is a whole token), the assumption used by our convergence proof (Assumption 4.4) carries over verbatim; empirically, tile-wise allocation reduces the quantization error at the same average bit budget.

### 3.3 Hadamard-Based Outlier Suppression Transform

Outliers in the activation values can severely degrade the accuracy of low-bit quantization even within a small quantization group. To mitigate quantization error caused by extreme outliers in activation groups, we propose an adaptive Hadamard transform strategy, which consists of three steps: (**i**) a *heuristic-based outlier detection* to decide if transform is needed, (**ii**) a *Hadamard transform with pivot element swap* to redistribute the outlier values in the quantization group, and (**iii**) an *asymmetric uniform quantization* of the values in the quantization group.

**Outlier detection heuristic**: Given any quantization group $\mathbf{a}_{i,j,t} = [a_1^{i,j,t}, a_2^{i,j,t}, \ldots, a_G^{i,j,t}] \in \mathbb{R}^G$, where $G = \frac{C}{N_t}$ is the quantization group size. For the rest parts in Section 3.3, we simplify the notation as $\mathbf{a}_{i,j,t} = \boldsymbol{\alpha} = [\alpha_1, \alpha_2, \ldots, \alpha_G] \in \mathbb{R}^G$ to introduce the quantization method within each tile. In order to detect whether an outlier is present, we define the following heuristic:

$$r = \frac{|\alpha^{(1)}|}{|\alpha^{(2)}| + \varrho} \tag{3}$$

Where $\alpha^{(1)}$ and $\alpha^{(2)}$ represent the elements in $\boldsymbol{\alpha}$ with the largest and the second largest absolute values, $\varrho$ is a small positive constant. If $r$ exceeds a threshold $\tau$ (empirically, we set $\tau = 2.0$), we will deem $\mathbf{a}_{i,j,t}$ to contain an outlier and apply the Hadamard-based transform as we will introduce below; otherwise, we skip this transform for that tile. We choose $\tau$ via a sensitivity study: $\tau = 2.0$ consistently improves convergence compared to always applying ($\tau = 0$) or never applying ($\tau = \infty$) the transform; detailed results are provided in Appendix G.

**Hadamard transform with pivot element swap**: For a group identified to have an outlier, we perform a *pivot element swap* to align the pivot (the element with the largest absolute value) with the Hadamard matrix structure. Let $d = \arg\max_k |\alpha_k|$ denote the pivot element index (i.e., $\alpha_d = \alpha^{(1)}$). We define a permutation matrix $\mathbf{P}_d \in \mathbb{R}^{G \times G}$ that swaps the first and $d$-th coordinates, which yields a permuted vector by multiplying this permutation matrix:

$$[\alpha_d, \alpha_2, \ldots, \alpha_1, \ldots, \alpha_G] = [\alpha_1, \alpha_2, \ldots, \alpha_G]\mathbf{P}_d = \boldsymbol{\alpha}\mathbf{P}_d$$

Next, we multiply this transformed vector by a Hadamard matrix $\mathbf{H}_G \in \{\pm 1\}^{G \times G}$ to redistribute the values and resolve the issue of outliers:

$$\dot{\boldsymbol{\alpha}} = \boldsymbol{\alpha}\mathbf{P}_d \frac{1}{\sqrt{G}}\mathbf{H}_G \tag{4}$$

After applying this transform, the extreme value in the original $\boldsymbol{\alpha}$ will be redistributed across all components in the transformed vector $\dot{\boldsymbol{\alpha}}$. This transform greatly reduces the dynamic range of the group: the formerly pivot value is no longer isolated in a single position, yielding a more balanced tile for the activation vector. As a result, the quantization error can be reduced, since a tighter quantization scale can represent the values with higher precision. Notably, because $\mathbf{H}_G$ is orthogonal (i.e., $\mathbf{H}_G\mathbf{H}_G^T = G\mathbf{I}_G$), we can later invert the transform by applying $\mathbf{H}_G^T$ to the de-quantized values when recovery of the original domain is required.

**Asymmetric uniform quantization**: After the above two steps, outlier-dominated tiles typically have a reduced dynamic range, making them more suitable for low-bit asymmetric uniform quantization. Thus, we can apply a standard asymmetric quantizer (i.e., You et al. (2024)) — if the computed heuristic $r \leq \tau$, we apply this quantizer for the original vector $\boldsymbol{\alpha}$; otherwise, we apply this quantizer for the transformed vector $\dot{\boldsymbol{\alpha}}$.

### 3.4 Overhead Analysis

Following the notation in Sections 3.1 and 3.2, we analyze the overhead of TAH-QUANT both theoretically and empirically. On the computation side, pivot swapping selects the top-2 entries per quantization tile and performs a conditional swap, costing $\mathcal{O}(G)$ per tile and $\mathcal{O}(BSN_tG)$ overall, implemented via parallel tensor operations. Entropy computation reduces over $A$ channels per allocation tile, giving $\mathcal{O}(BSC)$ in aggregate. Tile-wise bit allocation ranks $SM = S \cdot C/A$ entropy values per sample, costing $\mathcal{O}(B\,SM \log(SM))$, which remains negligible relative to attention and FFN in our evaluated settings. For outlier tiles, the Hadamard transform applies a $G \times G$ multiplication, costing $\mathcal{O}(G^2)$ per quantization tile and at most $\mathcal{O}(BSN_tG^2)$ in the worst case. On the communication side, under the default 80% `INT4` + 20% `INT3` setting with $A = G = 64$, TAH-QUANT transmits about 4.4 bits per activation element in the worst case. This includes roughly 0.625 bits of metadata for the per-tile scale, zero-point, the 1-bit precision selector, the 1-bit Hadamard/pivot transform flag, and the pivot index. A complete worst-case per-element bit accounting is given in Appendix E. These heterogeneous precisions and metadata are packed into a single contiguous `uint8` buffer sized for this worst case, so the communication layer observes a fixed-shape buffer rather than variable-length tensors. Thus, the transmitted budget is the reported worst-case value. This overhead is small compared with the dominant transformer computation, namely attention with complexity $\mathcal{O}(BSC(S + C))$ and FFN layers with complexity $\mathcal{O}(BSC^2)$. Empirically, profiling a 4-stage pipeline with micro-batch size 2 shows that TAH-QUANT adds only $\sim 1\%$ runtime overhead under the default $A = G = 64$ configuration.

### 3.5 TAH-Quant in Pipeline Parallel Training

Given the carefully designed TAH-QUANT quantization method, it is straightforward to integrate it into standard pipeline-parallel training. We illustrate this process in Algorithm 1. For clarity, we present a two-stage pipeline, which can be easily extended to an arbitrary number of stages. Following `AQ-SGD`, we use a simple fixed-point (naive) compressor for gradients in the backward pass. This choice is motivated by system efficiency rather than an algorithmic limitation. Backward propagation typically incurs substantially more computation than forward propagation, providing more opportunity for computation–communication overlap, so a higher-bit naive quantizer (e.g., 6–8 bits) is usually sufficient in practice; we illustrate this forward/backward compute–communication overlap schematically in Appendix L. We emphasize that TAH-QUANT also applies to backward gradients under ultra-low precision. An empirical comparison is provided in Appendix I. The complete per-tensor TAH-QUANT operator—entropy computation, top-$p\%$ selection, outlier detection, pivot swap, Hadamard transform, asymmetric quantization, and the exact packing and unpacking of payload and metadata—is specified step by step in Algorithms 2 and 3 (Appendix D).

---

**Algorithm 1** TAH-QUANT in a two-stage pipeline parallel training.

---

1: **Initialize:** sub-network $a(-)$ weights $\mathbf{x}^{(a)}$, sub-network $b(-)$ weights $\mathbf{x}^{(b)}$, optimizer $\rho$.
2: **for** t = 1, . . . , T **do**
3:      Randomly sample training batch $\xi_t$.
     // Forward propagation:
4:      Machine $a$ sends the quantized output activations $Q_{\text{TAH-QUANT}}\left(a(\xi_t, \mathbf{x}_t^{(a)})\right)$ to Machine $b$.
5:      Machine $b$ dequantizes the received activation $Q_{\text{TAH-QUANT}}\left(a(\xi_t, \mathbf{x}_t^{(a)})\right)$.
     // Backward propagation:
6:      Machine $b$ sends the quantized gradients w.r.t the activations $Q_{\text{NAIVE}}\left(\nabla_a(F \circ b)|_{\xi_t}\right)$ back to Machine $a$.
7:      Machine $a$ dequantizes the received gradient w.r.t the activations $Q_{\text{NAIVE}}\left(\nabla_a(F \circ b)|_{\xi_t}\right)$.
     // Parameter updates:
8:      Machine $a$ updates its parameters with gradients $\hat{\mathbf{g}}^t\left(\mathbf{x}^{(a)}\right)$ using optimizer $\rho$.
9:      Machine $b$ updates its parameters with gradients $\hat{\mathbf{g}}^t\left(\mathbf{x}^{(b)}\right)$ using optimizer $\rho$.
10: **end for**
11: **Output:** $\mathbf{x} = (\mathbf{x}_T^{(a)}, \mathbf{x}_T^{(b)})$

---

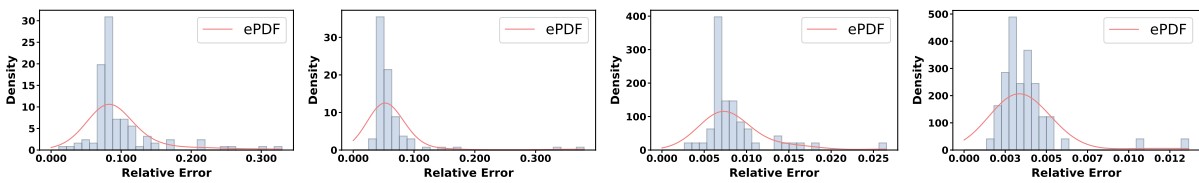

(a) step wise; tile size 64.   (b) step wise; tile size 32.   (c) full dataset; tile size 64.   (d) full dataset; tile size 32.

Figure 1: Empirical justification of Assumption 4.4.

## 4 Theoretical Analysis

In this section, we present convergence guarantees for TAH-QUANT, which aims to solve the following stochastic optimization problem in a pipeline-parallel fashion:

$$\min_{\mathbf{x}\in\mathbb{R}^d} \quad \mathbb{E}_{\xi\in\mathcal{D}}[F(\mathbf{x};\xi)] \tag{5}$$

where $\mathbf{x}$ denotes the model weights distributed across different pipelines, and $\xi$ represents random data drawn from the distribution $\mathcal{D}$. We denote $\nabla F(\mathbf{x};\xi)$ as the stochastic gradient and $\nabla f(\mathbf{x})$ as the full-batch gradient. Without loss of generality, we consider using momentum SGD as the optimizer $\rho$ in Algorithm 1:

$$\mathbf{m}^t = (1-\beta_1)\mathbf{m}^{t-1} + \beta_1\hat{\mathbf{g}}^t, \tag{6}$$
$$\mathbf{x}^{t+1} = \mathbf{x}^t - \eta\mathbf{m}^t, \tag{7}$$

where $\beta_1 \in (0,1)$ is the momentum coefficient and $\eta$ is the learning rate. The vector $\hat{\mathbf{g}}^t$ is a quantized estimate of the stochastic gradient $\nabla F(\mathbf{x}^t;\xi^t)$, obtained through Lines 3–7 of Algorithm 1. Specifically, it takes the form $\left(\hat{\mathbf{g}}^t(\mathbf{x}^{(a_1)}), \hat{\mathbf{g}}^t(\mathbf{x}^{(a_2)}), \ldots, \hat{\mathbf{g}}^t(\mathbf{x}^{(a_N)})\right)$, where $a_1, a_2, \ldots, a_N$ index the machines in the pipeline-parallel system. Our analysis extends to the Adam optimizer: Appendix N gives the full generalization (Theorem N.2), of which the momentum-SGD guarantee below is the special case where Adam's adaptive preconditioner is the identity, at the cost of one additional, standard assumption controlling the conditioning of that preconditioner. This Adam extension is a conditional, proof-level result with conservative constants: it reconciles the optimizer form rather than certifying every production hyperparameter used in large-scale LLM training.

### 4.1 Assumptions

**Assumption 4.1** (Lower Boundedness). The loss function $f : \mathbb{R}^d \to \mathbb{R}$ satisfies $\inf_{\mathbf{x}\in\mathbb{R}^d} f(\mathbf{x}) > -\infty$.

**Assumption 4.2** ($L$-Smoothness). The loss function $f$ is $L$-smooth, $i.e.$, it holds for any $\mathbf{x}, \mathbf{y} \in \mathbb{R}^d$ that

$$\|\nabla f(\mathbf{x}) - \nabla f(\mathbf{y})\|_2 \le L\|\mathbf{x} - \mathbf{y}\|_2.$$

**Assumption 4.3** (Stochastic Gradient). We assume that for some $\sigma > 0$, the stochastic gradient oracle satisfies

$$\mathbb{E}\left[\nabla F(x^t;\xi^t)\right] = \nabla f(x^t),$$
$$\mathbb{E}\left[\|\nabla F(x^t;\xi^t) - \nabla f(x^t)\|^2\right] \le \sigma^2. \tag{8}$$

Assumptions 4.1–4.3 are standard assumptions commonly used in stochastic optimization. The following assumption states that gradient quantization through TAH-QUANT proposed in Algorithm 1 does not introduce significant distortion to the true stochastic gradient.

**Assumption 4.4** (Quantization Error). Let $\mathbf{g}^t$ denote the original stochastic gradient $\nabla F(\mathbf{x}^t, \xi^t)$, and $\hat{\mathbf{g}}^t$ denote the quantized stochastic gradient obtained through TAH-QUANT. For some $\delta \in (0,1]$, it holds that

$$\|\hat{\mathbf{g}}^t - \mathbf{g}^t\|^2 \le (1-\delta)\|\mathbf{g}^t\|^2, \tag{9}$$
$$\|\mathbb{E}_{\xi^t\sim\mathcal{D}}[\hat{\mathbf{g}}^t] - \nabla f(\mathbf{x}^t)\|^2 \le (1-\delta)\|\nabla f(\mathbf{x}^t)\|^2, \tag{10}$$

The above assumption ensures that the quantized gradient $\hat{\mathbf{g}}$ remains close to the true gradient $\mathbf{g}$, with their closeness measured by the quantization coefficient $\delta$. A larger $\delta$ (*i.e.*, $\delta \to 1$) indicates a smaller quantization error. When $\delta = 1$, we have $\hat{\mathbf{g}} = \mathbf{g}$, implying no quantization error.

**Empirical justification of Assumption 4.4.** We now empirically verify that TAH-QUANT satisfies Assumption 4.4. To validate inequality (9), we conduct fine-tuning experiments on the Gemma2-2B model using the Math-7K dataset. At each training step, we compute the relative error $\|\hat{\mathbf{g}}^t - \mathbf{g}^t\|^2/\|\mathbf{g}^t\|^2$, as shown in Figures 1a and 1b. The results indicate that the relative errors remain below 0.4 across all steps, providing empirical support for (9) with $\delta = 0.6$. To validate inequality (10), we conduct experiments on the same model and dataset. At each step, we estimate the compressed-gradient expectation $\mathbb{E}_{\xi^t \sim \mathcal{D}}[\hat{\mathbf{g}}^t]$ by its empirical average over the dataset, together with the full-batch gradient $\nabla f(\mathbf{x}^t)$, and then evaluate the relative error $\|\mathbb{E}_{\xi^t \sim \mathcal{D}}[\hat{\mathbf{g}}^t] - \nabla f(\mathbf{x}^t)\|^2/\|\nabla f(\mathbf{x}^t)\|^2$, as shown in Figures 1c and 1d. All relative errors are below 0.1, so (10) holds with an even larger margin; since Assumption 4.4 uses a single $\delta$, the binding constraint is (9), and the assumption is empirically supported with $\delta = 0.6$ (inequality (10) alone would allow up to $\delta = 0.9$). In both experiments, we use tile sizes of 64 and 32, with 80% `INT4` and 20% `INT3` quantization. We further verify Assumption 4.4 on five models (`GPT-2XL`, `LLaMA-3.2-1B/3B`, `LLaMA-3.1-8B`, and `Qwen2.5-3B`, at the deployment PP=4 setting) in Appendix J (Table 12), where the relative errors remain well below 1. Complementary to this empirical check, Appendix M clarifies the theoretical status of Assumption 4.4. It derives activation-level error bounds from the TAH-QUANT design parameters and bounds the backward compressor's contribution. It further shows that, for nonlinear objectives, activation fidelity alone is insufficient to guarantee a uniform sample-wise relative bound on the gradient error. We therefore state the gradient-level contraction as an assumption, consistent with standard biased-compression analyses. These experiments demonstrate the effectiveness of TAH-QUANT, which quantizes variables to smaller sizes without incurring significant errors.

## 4.2 Convergence Guarantees

Under the above assumptions, we are ready to provide convergence guarantees of our proposed TAH-QUANT method.

**Theorem 4.5.** *Under Assumptions 4.1–4.4, if $\delta \in (0,1)$, $\beta_1 \in \left(0, \frac{\delta}{24 - 12\delta}\right)$ and $\eta \leq \min\left\{\frac{1}{2L}, \frac{\beta_1}{L} \cdot \sqrt{\frac{\delta}{8}}\right\}$, TAH-QUANT with momentum SGD converges as*

$$\frac{1}{T+1}\sum_{t=0}^{T}\mathbb{E}\Big[\big\|\nabla f(\mathbf{x}^t)\big\|_2^2\Big] \leq \frac{8\big(f(\mathbf{x}^0) - \inf_{\mathbf{x}} f(\mathbf{x})\big)}{\delta\,\eta\,(T+1)} + \frac{8\left\|\mathbf{m}^0 - \nabla f(\mathbf{x}^0)\right\|_2^2}{\delta\,\beta_1\,(T+1)}$$
$$+ \frac{24\,\beta_1\,\sigma^2}{\delta}.$$

**Corollary 4.6.** *Under Assumptions 4.1–4.4, if we choose $\beta_1 = \left(\frac{24}{\delta} + \sigma\sqrt{\frac{\delta^{1/2}(T+1)}{L\Delta}}\right)^{-1}$, $\eta = \left(2L + \frac{2^{3/2}L}{\delta^{1/2}\beta_1}\right)^{-1}$, TAH-QUANT with momentum SGD converges as*

$$\frac{\sum_{t=0}^{T}\mathbb{E}[\|\nabla f(\mathbf{x}^t)\|_2^2]}{T+1} = \mathcal{O}\left(\frac{L\Delta}{\delta^{5/2}(T+1)} + \sqrt{\frac{L\Delta\sigma^2}{\delta^{5/2}(T+1)}}\right),$$

*where $\Delta := f(\mathbf{x}^0) - \inf_{\mathbf{x}} f(\mathbf{x}) + (\delta/L) \cdot \|\mathbf{m}^0 - \nabla f(\mathbf{x}^0)\|_2^2$ (Proofs are in Appendix B).*

**Remark.** Corollary 4.6 yields three key implications. First, it guarantees that the proposed TAH-QUANT algorithm converges to a stationary solution of problem (5). Second, it shows that TAH-QUANT achieves a convergence rate of $\mathcal{O}(1/\sqrt{T})$, matching that of vanilla momentum SGD without gradient quantization. This demonstrates that TAH-QUANT effectively preserves the valuable gradient information during quantization. Third, the theorem indicates that the convergence rate is affected by the quantization error, quantified by the coefficient $\delta$. This is consistent with our expectations. Since TAH-QUANT maintains a relatively large $\delta$ (*i.e.*, close to 1), the quantization error remains moderate and does not significantly slow convergence.

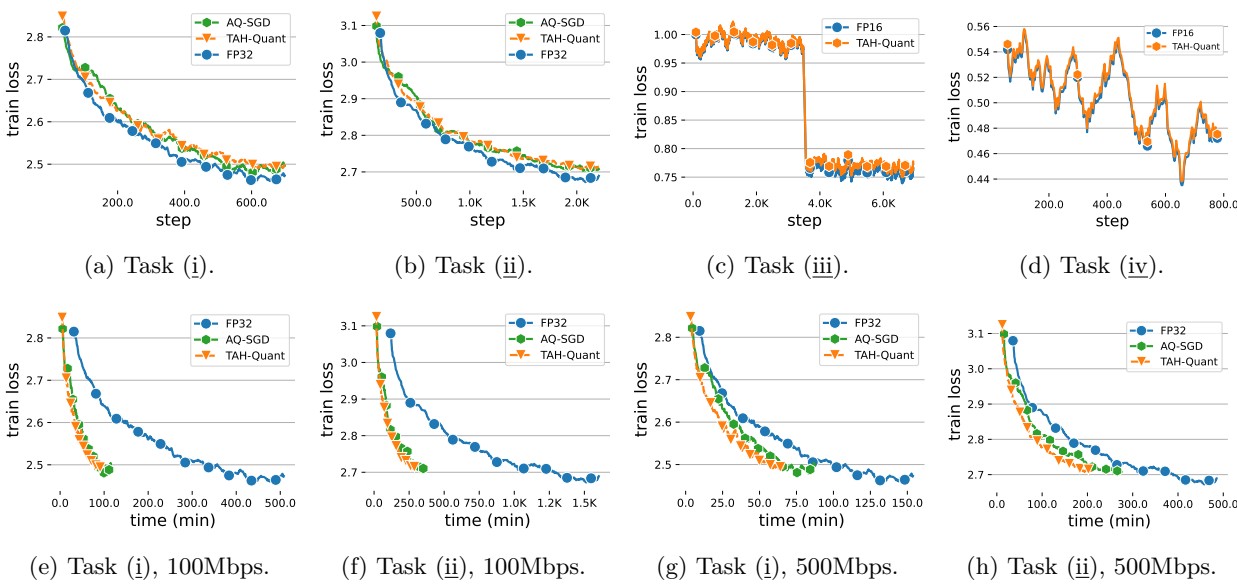

(a) Task (i).  (b) Task (ii).  (c) Task (iii).  (d) Task (iv).

(e) Task (i), 100Mbps. (f) Task (ii), 100Mbps. (g) Task (i), 500Mbps. (h) Task (ii), 500Mbps.

Figure 2: Top row: training-convergence comparison on the fine-tuning and instruction-tuning tasks (loss vs. steps). Bottom row: end-to-end training performance under different network bandwidths (loss vs. wall-clock time) on tasks (**i**) and (**ii**). See Section 5.1 for the detailed experimental setup of each task. Pretraining task (**v**) is summarized in Table 1. `AQ-SGD` is shown only on tasks (**i**)–(**ii**), where its activation cache is feasible; it is inapplicable to the instruction-tuning tasks (**iii**)–(**iv**) (prohibitive cache / single-epoch; Appendix Q).

## 5 Evaluation

We demonstrate that TAH-QUANT significantly accelerates LLM training over slow network connections. Specifically, we show that: (**i**) on representative benchmarks, TAH-QUANT enables aggressive quantization of activations and gradients without compromising convergence performance or incurring notable additional system overhead (Section 5.2); and (**ii**) the effectiveness of our system design is validated through a series of ablation studies (Section 5.3).

### 5.1 Experimental Setup

**Datasets and benchmarks.** We evaluate the proposed method on five training scenarios spanning language modeling and instruction-following tasks. Specifically, we fine-tune `GPT-2XL` (1.5B parameters) on (**i**) `WikiText-2`, a standard Wikipedia-based language modeling benchmark, and (**ii**) `ArXiv21`, a corpus of research paper abstracts from arXiv. To assess performance on instruction data, we fine-tune `Qwen2.5-3B` (3B parameters) on (**iii**) `Magicoder-Evol-Instruct-110K`, a dataset of 110k instruction-response pairs, and (**iv**) `Open-Platypus`, a composite open-source instruction tuning dataset covering multiple domains. Finally, we launch from-scratch pretraining of `LLaMA-3.2-1B` (1.2B parameters) on (**v**) `SlimPajama-6B`, a deduplicated 6B-token subset of the SlimPajama corpus tokenized with the LLaMA-3.2 tokenizer. These setups cover both general and specialized tasks, as well as supervised instruction tuning and LLM pretraining. We include a supplementary `LLaMA-8B` pretraining run on `Proof-Pile` in Appendix C. We use `GPT-2XL` for tasks (**i**)–(**ii**) to match the exact model and datasets of our baseline `AQ-SGD` Wang et al. (2022), enabling a direct comparison; the more recent `Qwen2.5-3B` and `LLaMA-3.2-1B` cover the instruction-tuning and pretraining settings.

**Distributed cluster.** The fine-tuning and instruction-tuning experiments (Tasks (**i**)–(**iv**)) are conducted on UCloud ucl using a distributed cluster of 8 instances, each equipped with an Nvidia RTX 3090 GPU. Each model is partitioned into 8 or 4 pipeline stages (one stage per GPU) to execute pipeline parallelism. The cluster's default interconnect bandwidth is 10 Gbps. To emulate slow-network conditions, we throttle inter-instance communication using Linux traffic control (tc), artificially limiting the bandwidth to sub-1 Gbps during training. This setting is common in decentralized environments and is frequently used in prior

LLM training evaluations Kim et al. (2025); Lim et al. (2024); Li et al. (2024); Erben et al. (2023); Wang et al. (2023b); Borzunov et al. (2023); similar constraints also arise in decentralized inference Mei et al. (2025).

**Pretraining setup (Task (v)).** Because the pretraining workload is substantially heavier than fine-tuning, we run Task (**v**) on a single 8-GPU server shared by both the `FP16` baseline and the `TAH-Quant` run, each occupying 4 GPUs and configured with pipeline depth PP=4. Sequence length is 4096, the global batch size is 256 (i.e. $\sim$1.05 M tokens / step), and the learning rate follows a cosine decay (peak $1.5\times10^{-4}$, min $1\times10^{-5}$).

**Baselines.** We compare our approach with two baseline communication strategies[1].

- `FP32/FP16`, which uses full-precision 32-bit (in Tasks (**i**) and (**ii**)) or 16-bit floating point (in Tasks (**iii**), (**iv**), and (**v**)) communication with no compression.
- `AQ-SGD`, the error-compensated low-bit activation quantization method with theoretical convergence guarantees Wang et al. (2022).

We additionally compare against a recent communication-efficient model-parallel method, Subspace Networks Ramasinghe et al. (2025), which reduces traffic by constraining the weight parameterization and is complementary to TAH-QUANT; the comparison is reported in Appendix R. We deliberately select baselines that compress the same quantity we target—inter-stage activation communication *during training*: `AQ-SGD` (the established error-compensated activation-compression method for this setting) and Subspace Networks. Inference-time weight-quantization methods such as QuaRot and SpinQuant (Section 2) are not directly comparable: they calibrate offline on fixed weights and do not affect gradient fidelity or convergence, whereas activation compression during training does. The necessity of each component of our own design is further isolated by the equal-bit ablation in Appendix P.

**Default bit allocation.** Unless otherwise specified, TAH-QUANT uses 80% INT4 + 20% INT3 mixed-precision activation quantization, with quantization tile size $G = 64$ and allocation-tile size $A = 64$. This 4/3-bit setting is the lowest-bit configuration that remains consistently stable in training; more aggressive choices (e.g., INT2) often lead to non-convergence. The 80/20 split balances communication reduction and convergence stability (Appendix H).

Table 1: Pretraining `LLaMA-3.2-1B` from scratch on `SlimPajama-6B` (PP=4, sequence length 4096, global batch size 256). Train loss is the rolling mean over a 300-step window; validation perplexity is measured at the closest evaluation checkpoint. Three-seed replicate evidence on `GPT-2XL` is provided in Appendix P.

| | *Train loss* ($\downarrow$) | | | | | | *Validation perplexity* ($\downarrow$) | | | | | |
| **Tokens** | 1 B | 2 B | 3 B | 4 B | 5 B | 6 B | 1 B | 2 B | 3 B | 4 B | 5 B | 6 B |
|---|---|---|---|---|---|---|---|---|---|---|---|---|
| `FP16` | 3.778 | 3.326 | 3.149 | 3.032 | 2.954 | 2.893 | 39.18 | 28.54 | 25.36 | 23.03 | 21.42 | 20.67 |
| `TAH-Quant` | 3.781 | 3.325 | 3.147 | 3.031 | 2.953 | 2.891 | 38.47 | 28.51 | 25.30 | 22.99 | 21.41 | 20.64 |

Table 2: `Qwen2.5-3B` SFT evaluation. Fine-tuning data are Open-Platypus for ARC/TQ/WG and Magicoder for HE; see Appendix A.

| Model | AVG | Open-Platypus | | | Magicoder |
|---|---|---|---|---|---|
| | | ARC | TQ | WG | HE |
| `Qwen` | 51.13 | 47.35 | 48.85 | 68.67 | 39.63 |
| `SFT-FP16` | 59.08 | 50.00 | 50.49 | 69.38 | 66.46 |
| `SFT-TAH` | 59.30 | 49.91 | 49.61 | 70.00 | 67.68 |
| `tile-INT4` | 55.93 | 47.27 | 48.60 | 68.43 | 59.42 |
| `tile-INT3` | 54.53 | 43.34 | 47.37 | 67.96 | 59.45 |
| `channel-INT4` | 44.13 | 36.52 | 43.02 | 61.01 | 35.95 |

---

[1]Each baseline is integrated into the same pipeline parallel training setup for fair comparison.

## 5.2 End-to-End Performance Results

To systematically evaluate the proposed TAH-QUANT quantization method, we conduct the experiment and report the results in terms of training convergence, downstream performance, and end-to-end training time.

**Convergence.** Figure 2 (top row) and Table 1 together summarize the convergence comparisons across tasks, demonstrating the efficacy and robustness of TAH-QUANT. Specifically, on tasks (**i**) and (**ii**), where `AQ-SGD` is executable due to manageable dataset sizes and a multi-epoch training paradigm, TAH-QUANT achieves comparable or slightly superior convergence compared to `AQ-SGD`. On larger-scale tasks (i.e., tasks (**iii**), (**iv**), and (**v**)), where `AQ-SGD` becomes infeasible due to prohibitive storage requirements (Task (**iii**)) or the single-epoch training constraint (Tasks (**iv**) and (**v**))—both quantified in Appendix Q—TAH-QUANT still closely matches the standard `FP16` baseline. For from-scratch pretraining of `LLaMA-3.2-1B` on `SlimPajama-6B` (Table 1), TAH-QUANT matches `FP16` not only in training loss but also in held-out validation perplexity throughout the entire ∼6 B-token training window, indicating that the proposed activation quantization does not degrade out-of-sample performance.[2] Furthermore, Table 2 reports downstream evaluations for the SFTed `Qwen2.5-3B` model in Tasks (**iii**) and (**iv**) across `FP16` and all feasible compression baselines: TAH-QUANT, fixed tile-wise `INT4`/`INT3` without the Hadamard transform (`tile-INT4`/`tile-INT3`), and a coarse per-channel `INT4` quantizer (`channel-INT4`). Fine-tuning with TAH-QUANT preserves model quality and matches `FP16` (59.30 vs. 59.08 average), while exceeding the equal-class `tile-INT4` baseline (55.93); the coarse `channel-INT4` quantizer degrades sharply (44.13), confirming that fine-grained tiling is necessary. Overall, the results highlight that TAH-QUANT enables aggressive activation quantization without extra memory overhead across the fine-tuning, instruction-tuning, and pretraining settings we evaluate.

**End-to-end training time.** TAH-QUANT is designed to preserve per-step convergence while reducing communication time, so its benefit appears primarily in wall-clock efficiency under bandwidth-limited networks rather than in fewer optimization steps. As illustrated in the bottom row of Figure 2, TAH-QUANT reaches the same loss faster than both uncompressed communication and `AQ-SGD`, with up to 1.33× wall-clock speedup over `AQ-SGD`. We attribute the speedup over `AQ-SGD` to eliminating the activation-cache offloading used by its error-compensation mechanism. Table 3 reports the corresponding `GPT-2XL` training throughput under 1Gbps,

Table 3: `GPT-2XL` throughput (tokens / s).

| Network Bandwidth | FP32 | AQ-SGD fw4 bw8 | TAH-Q fw~4 bw6 |
|---|---|---|---|
| 1Gbps | 2600 | 4749 | 5650 |
| 500Mbps | 2482 | 4311 | 5749 |
| 300Mbps | 1761 | 4369 | 5120 |
| 100Mbps | 751 | 3310 | 4045 |

500Mbps, 300Mbps, and 100Mbps bandwidth; TAH-QUANT achieves up to 5.4× throughput speedup over uncompressed `FP32`. Beyond bandwidth, we further evaluate TAH-QUANT under network latency, jitter, heterogeneous participants, and deeper pipelines (up to PP16 / 16 GPUs) in Appendix O, where the speedup grows as conditions degrade. Appendix F reports a sensitivity study over bandwidth and micro-batch size, showing that TAH-QUANT benefits most under lower bandwidth and moderate micro-batch sizes (mbs=2–4).

## 5.3 Ablation Study

To evaluate the contributions of each module in TAH-QUANT to reduce quantization error under pipeline parallel training, we perform a series of ablation studies and enumerate the experimental results below:

Table 4: Ablation: tile-wise quantization group size (TS).

| TS | MMLU | ARC |
|---|---|---|
| 8 | 64.60 | 50.34 |
| 32 | 64.88 | 49.91 |
| 64 | 64.80 | 49.49 |
| 128 | 64.34 | 49.66 |

First, to study how the **tile-wise quantization group size** influences statistical efficiency, we vary the group size to 8, 32, 64, and 128 and compare SFTed `Qwen2.5-3B` models over the set of benchmarks. Table 4 shows that MMLU is stable across the smaller tile sizes (TS=8/32/64, all within 64.6–64.9) and only degrades at TS=128 (64.34), while ARC differences across all tile sizes are under one point (49.49–50.34). We attribute the large-tile MMLU drop to the within-group statistical heterogeneity that grows with TS: as the group spans more channels, the single shared scale

---

[2]We further discuss backward-gradient quantization and numerical stability in Appendix I.

factor must accommodate increasingly disparate activation magnitudes, inflating the per-element quantization error. We adopt TS=64 as our default, which retains the accuracy while reducing the metadata overhead.

Second, to examine the effectiveness of the **entropy-guided adaptive bit allocation**, we compare TAH-QUANT with adaptive bit allocation enabled against a variant without adaptive allocation. Results in Figure 3a show that adaptive allocation accelerates training convergence — the training loss consistently decreases faster with adaptive bit allocation, reflecting reduced quantization error during compression. These observations validate our design of incorporating entropy-based tile-wise bit-width allocation.

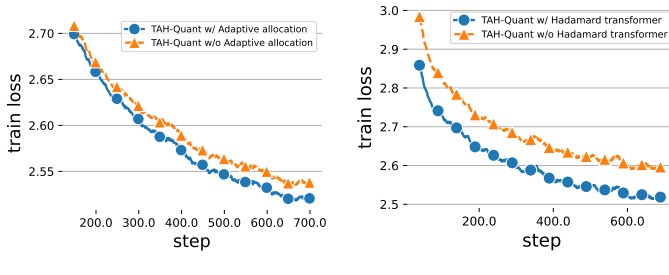

(a) Adaptive bit allocation.     (b) Hadamard transform.

Figure 3: Ablation studies on TAH-QUANT components.

Third, we evaluate the necessity of the **Hadamard-based outlier suppression** component in TAH-QUANT. Comparing training performance between setups with and without the Hadamard transform reveals that including this transform notably improves training stability and convergence speed — in Figure 3b, the training loss is substantially lower across the training phase when the Hadamard transform is applied, underscoring its effectiveness in mitigating quantization errors from outliers. This finding confirms the value of combining pivot element swapping with the Hadamard transform.

## 6 Conclusion

We present TAH-QUANT, a novel activation quantization method that alleviates communication bottlenecks in bandwidth-limited pipeline-parallel training of LLMs. TAH-QUANT integrates fine-grained tile-wise quantization for localized error control, entropy-guided tile-wise bit allocation that refines token-level allocation, and a Hadamard-based transform with pivot swapping to mitigate outliers. Under an empirically verified bounded relative quantization-error condition, we show that pipeline-parallel training with TAH-QUANT preserves the same convergence rate $(\mathcal{O}(1/\sqrt{T}))$ as standard SGD. Empirical results demonstrate that TAH-QUANT compresses activations to 3–4 bits without degrading convergence, while improving throughput over uncompressed communication, reducing wall-clock time relative to AQ-SGD, avoiding activation-cache overhead, and maintaining consistent quality across the fine-tuning, instruction-tuning, and from-scratch pretraining settings we evaluate.

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

## A  Experimental Details

**Downstream evaluation protocol (Table 2).** We evaluate the SFTed `Qwen2.5-3B` models on four standard benchmarks: ARC Clark et al. (2018), TruthfulQA (TQ) Lin et al. (2022), WinoGrande (WG) Sakaguchi et al. (2021), and HumanEval (HE) Chen et al. (2021b). All evaluations are conducted in a zero-shot setting using the lm-evaluation-harness Gao et al. (2024) framework. We report normalized accuracy for ARC-Challenge, accuracy for WinoGrande, mc2 for TruthfulQA, and pass@1 for HumanEval. **Fine-tuning.** We fine-tune the `GPT-2XL` on `WikiText-2` and `ArXiv21` for 10 epochs. Specifically, we set the learning rate to 5.0e-6, the batch size to 32, and the micro-batch size to 1, max sequence length to 1024 for both datasets. The learning rate decays linearly after the warm-up stage.

**Instruction-tuning.** We perform instruction tuning on `Qwen2.5-3B` using `Open-Platypus` and `Magicoder-110K` for 1 and 2 epochs, respectively. The learning rate is set to 2.0e-5, with a batch size of 32 for both datasets. We use a cosine learning rate scheduler for `Open-Platypus`, and a cosine scheduler with a minimum learning rate of 2.0e-6 for `Magicoder-110K`.

**Pretraining (Task (v)).** We pretrain `LLaMA-3.2-1B` from scratch on the `SlimPajama-6B` corpus (a deduplicated 6 B-token subset of SlimPajama) tokenized with the LLaMA-3.2 tokenizer. The model has 16 transformer layers, hidden size 2048, 32 attention heads, 8 KV heads, and FFN hidden size 8192; we use sequence length 4096 and a global batch size of 256 (i.e. $\sim$1.05 M tokens / step). The peak learning rate is $1.5 \times 10^{-4}$ with cosine decay to a minimum of $1 \times 10^{-5}$, weight decay 0.1, $\beta_1$=0.9, $\beta_2$=0.95, gradient clipping at 1.0, and a small linear warmup. Pipeline depth is PP=4 (one transformer stage per GPU). The `FP16` baseline and the `TAH-Quant` arm run concurrently on the same 8-GPU server (4 GPUs each), sharing the same hardware and wall-clock budget. The main text reports training through $\sim$6 B tokens, where both arms reach the same loss and validation perplexity within noise.

A supplementary `LLaMA-8B` pretraining run on `Proof-Pile` is reported in Appendix C: 20$k$ iterations at global batch size 131,072 tokens (i.e. $\sim$2.5 B tokens, >30% of the corpus), peak learning rate $1.5 \times 10^{-4}$ with cosine decay to $1 \times 10^{-5}$, and weight decay 0.1.

## B  Complete Proofs

In this section, we provide detailed proofs for Theorem 4.5. We first prove the following lemma.

**Lemma B.1** (Descent lemma). *Under Assumption 4.2 and the update rule 7, it holds that*

$$f(\mathbf{x}^{t+1}) \leq f(\mathbf{x}^t) - \frac{\eta}{2}\|\nabla f(\mathbf{x}^t)\|_2^2 - \left(\frac{1}{2\eta} - \frac{L}{2}\right)\|\mathbf{x}^{t+1} - \mathbf{x}^t\|_2^2 + \frac{\eta}{2}\|\mathbf{m}^t - \nabla f(\mathbf{x}^t)\|_2^2. \tag{11}$$

*Proof.* by Assumption 4.2 we have

$$f(\mathbf{x}^{t+1}) \leq f(\mathbf{x}^t) + \eta\langle\nabla f(\mathbf{x}^t), \frac{1}{\eta}(\mathbf{x}^{t+1} - \mathbf{x}^t)\rangle + \frac{L}{2}\|\mathbf{x}^{t+1} - \mathbf{x}^t\|_2^2$$

$$= f(\mathbf{x}^t) - \frac{\eta}{2}\|\nabla f(\mathbf{x}^t)\|_2^2 - \frac{1}{2\eta}\|\mathbf{x}^{t+1} - \mathbf{x}^t\|_2^2 + \frac{\eta}{2}\|\nabla f(\mathbf{x}^t) - \mathbf{m}^t\|_2^2 + \frac{L}{2}\|\mathbf{x}^{t+1} - \mathbf{x}^t\|_2^2. \tag{12}$$

where the second equality uses $2\langle a, b\rangle = \|a\|_2^2 + \|b\|_2^2 - \|a - b\|_2^2$ □

**Lemma B.2** (momentum contraction). *Under Assumptions 4.1–4.4, if $\delta \in (0, 1)$, it holds that*

$$\mathbb{E}[\|\mathbf{m}^t - \nabla f(\mathbf{x}^t)\|_2^2] \leq \left(1 - \beta_1\left(1 - \frac{\delta}{2}\right)\right)\mathbb{E}[\|\mathbf{m}^{t-1} - \nabla f(\mathbf{x}^{t-1})\|_2^2] + \frac{2L^2}{\delta\beta_1}\mathbb{E}[\|\mathbf{x}^t - \mathbf{x}^{t-1}\|_2^2]$$
$$+ (\beta_1 + 6\beta_1^2)(1 - \delta)\mathbb{E}[\|\nabla f(\mathbf{x}^t)\|_2^2] + 3(2 - \delta)\beta_1^2\sigma^2. \tag{13}$$

*Proof.* According to the update of momentum 7, we have

$$\mathbf{m}^t - \nabla f(\mathbf{x}^t) = (1 - \beta_1)(\mathbf{m}^{t-1} - \nabla f(\mathbf{x}^{t-1}) + \nabla f(\mathbf{x}^{t-1}) - \nabla f(\mathbf{x}^t)) + \beta_1(\hat{\mathbf{g}}^t - \nabla f(\mathbf{x}^t)).$$

Taking expectation we have

$$
\begin{aligned}
\mathbb{E}[\|\mathbf{m}^t - \nabla f(\mathbf{x}^t)\|_2^2] =& \mathbb{E}[\|(1-\beta_1)(\mathbf{m}^{t-1} - \nabla f(\mathbf{x}^{t-1}) + \nabla f(\mathbf{x}^{t-1}) - \nabla f(\mathbf{x}^t)) + \beta_1(\mathbb{E}[\hat{\mathbf{g}}^t] - \nabla f(\mathbf{x}^t))\|_2^2] \\
& + \beta_1^2 \mathbb{E}[\|\hat{\mathbf{g}}^t - \mathbb{E}[\hat{\mathbf{g}}^t]\|_2^2].
\end{aligned}
\tag{14}
$$

For the first term, applying Jensen's inequality yields

$$
\begin{aligned}
& \mathbb{E}[\|(1-\beta_1)(\mathbf{m}^{t-1} - \nabla f(\mathbf{x}^{t-1}) + \nabla f(\mathbf{x}^{t-1}) - \nabla f(\mathbf{x}^t) + \beta_1(\mathbb{E}[\hat{\mathbf{g}}^t] - \nabla f(\mathbf{x}^t))\|_2^2] \\
\leq & (1-\beta_1)\mathbb{E}[\|\mathbf{m}^{t-1} - \nabla f(\mathbf{x}^{t-1}) + \nabla f(\mathbf{x}^{t-1}) - \nabla f(\mathbf{x}^t)\|_2^2] + \beta_1 \mathbb{E}[\|\mathbb{E}[\hat{\mathbf{g}}^t] - \nabla f(\mathbf{x}^t)\|_2^2].
\end{aligned}
\tag{15}
$$

By Young's inequality, we have

$$
\begin{aligned}
\mathbb{E}[\|\mathbf{m}^{t-1} - \nabla f(\mathbf{x}^{t-1}) + \nabla f(\mathbf{x}^{t-1}) - \nabla f(\mathbf{x}^t)\|_2^2] \leq & \left(1 + \frac{\delta\beta_1}{2}\right)\mathbb{E}[\|\mathbf{m}^{t-1} - \nabla f(\mathbf{x}^{t-1})\|_2^2] \\
& + \left(1 + \frac{2}{\delta\beta_1}\right)\mathbb{E}[\|\nabla f(\mathbf{x}^t) - \nabla f(\mathbf{x}^{t-1})\|_2^2].
\end{aligned}
\tag{16}
$$

For the second term, applying Cauchy's inequality yields

$$
\begin{aligned}
\mathbb{E}[\|\hat{\mathbf{g}}^t - \mathbb{E}[\hat{\mathbf{g}}^t]\|_2^2] \leq & 3\mathbb{E}\|\hat{\mathbf{g}}^t - \mathbf{g}^t\|_2^2 + 3\mathbb{E}[\|\mathbf{g}^t - \nabla f(\mathbf{x}^t)\|_2^2] + 3\mathbb{E}[\|\nabla f(\mathbf{x}^t) - \mathbb{E}[\hat{\mathbf{g}}^t]\|_2^2] \\
\leq & 3(1-\delta)\mathbb{E}[\|\nabla f(\mathbf{x}^t)\|_2^2] + 3(1-\delta)\mathbb{E}[\|\mathbf{g}^t\|_2^2] + 3\sigma^2, \\
\leq & 6(1-\delta)\mathbb{E}[\|\nabla f(\mathbf{x}^t)\|_2^2] + 3(2-\delta)\sigma^2,
\end{aligned}
\tag{17}
$$

where the inequality uses Assumption 4.3 and 4.4. Applying (15)(16)(17) to (14) and using Assumption 4.2 and 4.4, we obtain B.2. □

**Remark.** From this proof, it is evident that both inequalities in Assumption 4.4 are necessary. In particular, the second inequality is essential for bounding the variance of $\hat{\mathbf{g}}^t$, which plays a crucial role in the overall convergence analysis.

Now we are ready to prove Theorem 4.5. We first restate the theorem in Theorem B.3.

**Theorem B.3.** *Under Assumptions 4.1–4.4, if $\beta_1 \in (0, \delta/(24-12\delta))$, $\delta \in (0,1)$ and $\eta \leq \min\{1/2L, \sqrt{(\delta\beta_1^2)/(8L^2)}\}$, TAH-QUANT with momentum SGD converges as*

$$
\frac{1}{T+1}\sum_{t=0}^{T}\mathbb{E}[\|\nabla f(\mathbf{x}^t)\|_2^2] \leq \frac{8[f(\mathbf{x}^0) - \inf_{\mathbf{x}} f(\mathbf{x})]}{\delta\eta(T+1)} + \frac{8\|\mathbf{m}^0 - \nabla f(\mathbf{x}^0)\|_2^2}{\delta\beta_1(T+1)} + \frac{24\beta_1\sigma^2}{\delta}.
\tag{18}
$$

*Proof.* By Lemma B.1, we have

$$
f(\mathbf{x}^{t+1}) - f(\mathbf{x}^t) \leq -\left(\frac{1}{2\eta} - \frac{L}{2}\right)\|\mathbf{x}^{t+1} - \mathbf{x}^t\|_2^2 + \frac{\eta}{2}\|\nabla f(\mathbf{x}^t) - \mathbf{m}^t\|_2^2 - \frac{\eta}{2}\|\nabla f(\mathbf{x}^t)\|_2^2.
\tag{19}
$$

Taking expectation and summing (19) for $t = 0, 1, \cdots, T$ yields

$$
\begin{aligned}
\inf_{\mathbf{x}} f(\mathbf{x}) - f(\mathbf{x}^0) \leq & \frac{\eta}{2}\sum_{t=0}^{T}\mathbb{E}[\|\nabla f(\mathbf{x}^t) - \mathbf{m}^t\|_2^2] - \left(\frac{1}{2\eta} - \frac{L}{2}\right)\sum_{t=0}^{T}\mathbb{E}[\|\mathbf{x}^{t+1} - \mathbf{x}^t\|_2^2] \\
& - \frac{\eta}{2}\sum_{t=0}^{T}\mathbb{E}[\|\nabla f(\mathbf{x}^t)\|_2^2].
\end{aligned}
\tag{20}
$$

summing the inequality in Lemma B.2 for $t = 1, 2, \cdots, T$ we have

$$
\begin{aligned}
\beta_1\left(1 - \frac{\delta}{2}\right)\sum_{t=0}^{T}\mathbb{E}[\|\mathbf{m}^t - \nabla f(\mathbf{x}^t)\|_2^2] \leq & \|\mathbf{m}^0 - \nabla f(\mathbf{x}^0)\|_2^2 + \frac{2L^2}{\delta\beta_1}\sum_{t=1}^{T}\|\mathbf{x}^t - \mathbf{x}^{t-1}\|_2^2 \\
& + (1-\delta)(\beta_1 + 6\beta_1^2)\sum_{t=1}^{T}\mathbb{E}[\|\nabla f(\mathbf{x}^t)\|_2^2] + 3T(2-\delta)\beta_1^2\sigma^2.
\end{aligned}
\tag{21}
$$

noting that $\delta \in (0, 1)$ we obtain

$$\sum_{t=0}^{T} \mathbb{E}[\|\mathbf{m}^t - \nabla f(\mathbf{x}^t)\|_2^2] \leq \frac{2\|\mathbf{m}^0 - \nabla f(\mathbf{x}^0)\|_2^2}{\beta_1} + \frac{4L^2}{\delta\beta_1^2} \sum_{t=1}^{T} \|\mathbf{x}^t - \mathbf{x}^{t-1}\|_2^2$$

$$+ \left(1 - \frac{\delta}{2}\right)(1 + 6\beta_1) \sum_{t=1}^{T} \mathbb{E}[\|\nabla f(\mathbf{x}^t)\|_2^2] + 6T\beta_1\sigma^2. \quad (22)$$

Applying 22 to (20) and noting that $\beta_1 \in (0, \delta/(24 - 12\delta))$ implies $(1 - \delta/2)(1 + 6\beta_1) \leq 1 - \delta/4$, we obtain

$$\frac{1}{T+1} \sum_{t=0}^{T} \mathbb{E}[\|\nabla f(\mathbf{x}^t)\|_2^2] \leq \frac{8[f(\mathbf{x}^0) - \inf_{\mathbf{x}} f(\mathbf{x})]}{\delta\eta(T+1)} + \frac{8\|\mathbf{m}^0 - \nabla f(\mathbf{x}^0)\|_2^2}{\delta\beta_1(T+1)} + \frac{24\beta_1\sigma^2}{\delta}$$

$$- \frac{8}{\delta\eta}\left(\frac{1}{2\eta} - \frac{L}{2} - \frac{2\eta L^2}{\delta\beta_1^2}\right) \sum_{t=0}^{T} \|\mathbf{x}^{t+1} - \mathbf{x}^t\|_2^2. \quad (23)$$

Since $\eta \leq \min\{1/2L, \sqrt{(\delta\beta_1^2)/(8L^2)}\}$ implies $1/(4\eta) \geq L/2$ and $1/(4\eta) \geq (2\eta L^2)/(\delta\beta_1^2)$, (18) is a direct result of (23). □

## C    Supplementary Pretraining: `LLaMA-8B` on `Proof-Pile`

For completeness we also include a from-scratch pretraining run of `LLaMA-8B` on the `Proof-Pile` mathematical corpus, using the configuration described in the "Pretraining" paragraph of Appendix A ($20k$ iterations, global batch size 131,072 tokens, $\sim$2.5 B tokens total, peak learning rate $1.5{\times}10^{-4}$ with cosine decay, weight decay 0.1, PP=4).

We present this larger-model run as a supplementary scale check and use the smaller `LLaMA-3.2-1B` run as the main-text pretraining benchmark. The reason is statistical efficiency rather than model size alone: under Chinchilla-style scaling laws Hoffmann et al. (2022), the `LLaMA-3.2-1B` run receives substantially more tokens per parameter ($\sim$5) than the `LLaMA-8B` run ($\sim$0.3), making it the more informative setting for detecting mature pretraining-quality differences under the available token budget.

Figure 4 reports the training loss and the held-out validation perplexity for this run. The training-loss curve (Figure 4a) covers the full $\sim$0.4–2.5 B-token horizon after the initial warmup transient is dropped (matching the slicing convention used in Wang et al. (2022)). The validation-PPL curve (Figure 4b) shows perplexity at 0.5 B-spaced checkpoints over the 0.5–2.5 B-token range. Across the entire horizon, TAH-QUANT is essentially indistinguishable from the uncompressed `FP16` baseline: the validation-PPL gap stays within $\pm$0.013 at every measured checkpoint and shows no upward drift, indicating that the per-step quantization error of TAH-QUANT does not visibly accumulate even at the 8B-parameter scale.

## D    Complete TAH-Quant Operator: Pseudocode

Algorithm 1 shows how TAH-QUANT is embedded in the pipeline loop at a high level. We give the full per-tensor TAH-QUANT operator below: entropy computation, top-$p\%$ selection, outlier detection, pivot swap, Hadamard transform, asymmetric quantization, the exact byte packing of payload and *all* metadata, and the matching inverse on the receiver. Algorithm 2 (sender) and Algorithm 3 (receiver) mirror the reference implementation one-to-one (`compress`+`pack_data` and `unpack_data`+`decompress`). Here $G$ is the quantization-group (tile) size, $A$ the allocation-tile width ($A{=}C$ recovers token-level allocation), $(b_H, b_L){=}(4, 3)$ the high/low bit-widths, $p$ the high-precision fraction, $\tau$ the outlier threshold, and $\mathbf{H} \in \{\pm 1\}^{G \times G}$ the unnormalized Hadamard matrix.

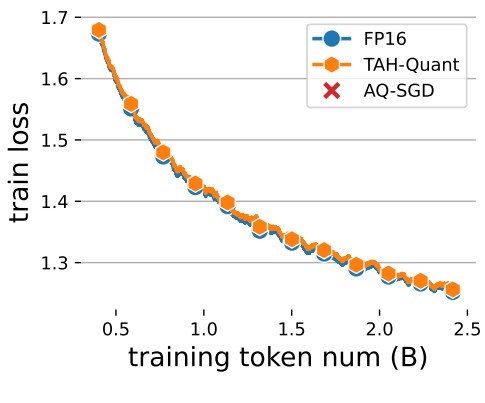
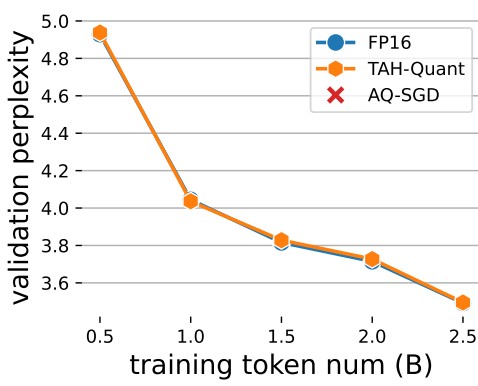

(a) Training loss vs. tokens.

(b) Validation perplexity vs. tokens.

Figure 4: Pretraining `LLaMA-8B` from scratch on `Proof-Pile` (PP=4, global batch size 32, sequence length 4096). TAH-QUANT (orange) tracks the uncompressed `FP16` baseline (blue) on both training loss (a) and held-out validation perplexity (b). `AQ-SGD` is shown in the legend for reference but is infeasible on this single-epoch pretraining workload (its activation cache would require storing per-microbatch forward activations for the entire corpus).

---

**Algorithm 2** TAH-QUANT operator — sender (compress & pack). Mirrors `compress` + `pack_data`.

---

1: **Input:** activation $\mathbf{a} \in \mathbb{R}^{B \times S \times C}$; $G$, $A$, $(b_H, b_L)$, $p$, $\tau$, $\mathbf{H}$.
2: Reshape the channel axis into allocation tiles of width $A$ (a no-op if $A=C$).
3: **// Stage 1: entropy-guided precision allocation**
4: **for all** allocation tiles $\tilde{\mathbf{a}}$ **do**
5: $\quad p_k \leftarrow |\tilde{a}_k|/(\|\tilde{\mathbf{a}}\|_1 + \epsilon); \quad \mathcal{H} \leftarrow -\sum_k p_k \log(p_k + \varsigma)$
6: **end for**
7: Rank the tiles of each sample by $\mathcal{H}$; assign $b_H$ (`INT4`) to the top-$p$ fraction (highest entropy) and $b_L$ (`INT3`) to the rest.
8: Record the 1-bit precision map $\mathbf{m}^{\text{tok}}$ and reorder tiles so low-precision tiles precede high-precision ones (the permutation is recoverable from $\mathbf{m}^{\text{tok}}$).
9: **// Stage 2: outlier detection + conditional transform**
10: Reshape into quantization tiles $\mathbf{t} \in \mathbb{R}^G$.
11: **for all** tiles $\mathbf{t}$ **do**
12: $\quad (v_1, v_2) \leftarrow$ top-2 of $|\mathbf{t}|; \quad f \leftarrow 1$ if $v_1 \geq \tau\, v_2$ else 0; $\quad$ store $f$ in $\mathbf{m}^{\text{tile}}$
13: $\quad$ **if** $f = 1$ **then**
14: $\quad\quad j^\star \leftarrow \arg\max_k |t_k|$ (store $j^\star$ in the pivot-index field $\Pi$); swap $t_0 \leftrightarrow t_{j^\star}$
15: $\quad\quad \mathbf{t} \leftarrow \mathbf{H}\mathbf{t}/\sqrt{G} \quad$ (Hadamard transform)
16: $\quad$ **end if**
17: **end for**
18: **// Stage 3: asymmetric per-tile quantization**
19: **for all** tiles $\mathbf{t}$ with bit-width $b \in \{b_H, b_L\}$ **do**
20: $\quad z \leftarrow \min(\mathbf{t}); \quad \mathbf{t} \leftarrow \mathbf{t} - z; \quad s \leftarrow \max(\mathbf{t})/(2^b - 1); \quad \mathbf{q} \leftarrow \text{clip}(\text{round}(\mathbf{t}/s), 0, 2^b - 1)$
21: **end for**
22: **// Stage 4: pack one contiguous `uint8` buffer (fixed order)**
23: Append, in order: low-precision payload ($b_L$ bits/elem), high-precision payload ($b_H$ bits/elem), zero-points $z$ and scales $s$ (`bf16`, 16 bits each), $\mathbf{m}^{\text{tok}}$ (1 bit/tile), $\mathbf{m}^{\text{tile}}$ (1 bit/tile), a fixed pivot-index field $\Pi$ ($\lceil \log_2 G \rceil$ bits per quantization tile; a dummy value for unflagged tiles); pad to a byte boundary.
24: **Output:** byte buffer $\mathbf{y}$ (fixed shape; contains everything the receiver needs).

---

---

**Algorithm 3** TAH-QUANT operator — receiver (unpack & decompress).  Mirrors `unpack_data` + `decompress`.

---
1: **Input:** byte buffer $\mathbf{y}$; tensor shapes; $G$, $A$, $(b_H, b_L)$, $p$, $\mathbf{H}$.
2: **// Unpack in exactly the order used when packing**
3: Read the low-/high-precision payload $\mathbf{q}$, then zero-points $z$ and scales $s$ (`bf16`→`fp32`), then $\mathbf{m}^{\mathrm{tok}}$, $\mathbf{m}^{\mathrm{tile}}$, and the fixed pivot-index field $\Pi$ (one slot per quantization tile; entries are used only for tiles with $\mathbf{m}^{\mathrm{tile}}{=}1$).
4: **// Dequantize**
5: **for all** tiles **do**
6: $\quad \mathbf{t} \leftarrow \mathbf{q} \cdot s + z$
7: **end for**
8: **// Inverse transform on flagged tiles (reverse of the sender's order)**
9: **for all** tiles with $\mathbf{m}^{\mathrm{tile}}{=}1$ **do**
10: $\quad \mathbf{t} \leftarrow \mathbf{H}^{\top}\mathbf{t}/\sqrt{G}$    (inverse Hadamard)
11: $\quad$ swap $t_0 \leftrightarrow t_{j^\star}$ using $\Pi$    (inverse pivot swap)
12: **end for**
13: **// Restore the original tile order**
14: Invert the Stage-1 permutation using $\mathbf{m}^{\mathrm{tok}}$ and reshape to $\hat{\mathbf{a}} \in \mathbb{R}^{B \times S \times C}$.
15: **Output:** dequantized activation $\hat{\mathbf{a}}$.

---

# E   Metadata and Transmitted Bit Budget

Table 5 gives the complete per-element bit accounting for the transmitted buffer under the default configuration ($A = G = 64$, 80% `INT4` + 20% `INT3`). Every field the receiver needs to dequantize and invert the transform is included, namely the per-tile scale and zero-point, the 1-bit precision selector, the 1-bit Hadamard/pivot transform flag, and the pivot index of each transformed tile. The pivot-index cost is counted at its *worst case*— every tile is Hadamard-transformed and therefore carries a pivot index—so the reported $\approx 4.43$ bit/element is the worst-case transmitted budget. To keep the communication layer simple, these heterogeneous fields are packed into a single contiguous, fixed-shape `uint8` buffer sized to exactly this worst case, so the transmitted budget is this worst-case value. (Only the flagged outlier tiles logically require a pivot index, so the underlying *information content* is lower, but the fixed-size buffer does not exploit this and we report the worst case.) The packed stream is rounded up to a whole number of bytes, which adds at most 7 bits to the entire buffer and is negligible per element.

Table 5: Per-element transmitted bit budget of TAH-QUANT at $A = G = 64$ with the default 80/20 `INT4`/`INT3` allocation.

| Component | bits / element |
|---|:---:|
| `INT4`/`INT3` payload ($0.8 \cdot 4 + 0.2 \cdot 3$) | 3.80 |
| per-tile scale (`bf16`, $16/G$) | 0.25 |
| per-tile zero-point (`bf16`, $16/G$) | 0.25 |
| precision selector ($1/G$) | 0.016 |
| Hadamard/pivot transform flag ($1/G$) | 0.016 |
| pivot index, worst case ($\lceil \log_2 G \rceil / G = 6/64$) | 0.094 |
| **Total** | $\approx \mathbf{4.43}$ |

# F   System Sensitivity Study: Bandwidth and Micro-batch Size

We fix the global batch size to 32 and sweep the micro-batch size (mbs) and inter-stage bandwidth. Tables 6–7 report throughput (tokens/s) and speedup.

Table 6: Throughput and speedup under 500 Mbps (global batch = 32).

| tokens / s | w/o TAH-QUANT | w/ TAH-QUANT | Speedup |
|---|---|---|---|
| mbs = 1 | 2406 | 4289 | 1.78× |
| mbs = 2 | 1710 | 4016 | 2.35× |
| mbs = 4 | 1317 | 3023 | 2.30× |
| mbs = 8 | 981 | 2099 | 2.14× |

Table 7: Throughput and speedup under 1 Gbps (global batch = 32).

| tokens / s | w/o TAH-QUANT | w/ TAH-QUANT | Speedup |
|---|---|---|---|
| mbs = 1 | 3449 | 5177 | 1.50× |
| mbs = 2 | 2576 | 4655 | 1.81× |
| mbs = 4 | 2003 | 3475 | 1.73× |
| mbs = 8 | 1485 | 2436 | 1.64× |

**Discussion.** TAH-QUANT yields larger speedups under lower bandwidth since inter-stage communication dominates (up to 2.35× at 500 Mbps). Speedup increases with micro-batch size and peaks at mbs=2–4, where communicated activation volume per step is larger. At a very large micro-batch size (mbs=8), increased pipeline idle time slightly reduces the speedup.

## G  Sensitivity of Outlier Threshold $\tau$

We evaluate the robustness of the outlier detection threshold $\tau$ in Eq. 3 by varying $\tau$. $\tau = 0$ corresponds to always applying the transform (treating every tile as containing outliers), while $\tau = \infty$ corresponds to never applying it. Table 8 shows that $\tau = 2.0$ yields the most stable and fastest loss decrease.

| Loss | Step 100 | Step 200 | Step 300 | Step 400 | Step 500 |
|---|---|---|---|---|---|
| $\tau = 0$ (always transform) | 2.97 | 2.79 | 2.63 | 2.60 | 2.60 |
| $\tau = 2$ (ours) | **2.72** | **2.63** | **2.59** | **2.56** | **2.55** |
| $\tau = 4$ | 2.73 | 2.65 | 2.61 | 2.58 | 2.57 |
| $\tau = \infty$ (never transform) | 2.82 | 2.72 | 2.68 | 2.64 | 2.63 |

Table 8: Effect of the outlier detection threshold $\tau$ on training loss.

## H  Additional Bit-Allocation Studies

### H.1  Varying INT4/INT3 Ratios

Table 9 reports training losses under different INT4/INT3 splitting ratios. The mixed setting lies between the two extremes, and increasing the INT4 fraction yields more stable convergence in this sweep.

| Loss | step 50 | step 100 | step 150 | step 200 | step 300 | step 400 | step 500 | step 600 |
|---|---|---|---|---|---|---|---|---|
| 100% INT4 | 2.78 | 2.70 | 2.66 | 2.61 | 2.56 | 2.52 | 2.51 | 2.49 |
| 50% INT4 + 50% INT3 | 2.82 | 2.74 | 2.71 | 2.66 | 2.62 | 2.58 | 2.57 | 2.55 |
| 100% INT3 | 2.85 | 2.77 | 2.74 | 2.69 | 2.65 | 2.62 | 2.60 | 2.58 |

Table 9: Loss trajectories under different INT4/INT3 ratio settings.

## H.2 Extremely Low-bit Configuration with INT2

We additionally evaluate a more aggressive configuration involving INT2. Table 10 shows that the INT2-involved setting exhibits degraded behavior in this experiment, consistent with instability under extremely low-bit quantization.

| Loss | step 0 | step 100 | step 200 | step 300 | step 400 | step 500 | step 600 |
|---|---|---|---|---|---|---|---|
| 80% INT2 + 20% INT3 | 3.31 | 3.22 | 3.20 | 3.19 | 3.17 | 3.18 | 3.15 |

Table 10: Loss trajectories with INT2-involved quantization, where training becomes unstable.

# I  Backward Gradients under Ultra-low Precision

In the main text, we quantize backward gradients with a higher-bit fixed-point compressor to strike a practical balance between system efficiency and numerical stability. Here we stress-test backward quantization under an ultra-low precision budget (4-bit). As shown in Table 11, the naive 4-bit fixed-point scheme diverges, whereas applying TAH-QUANT to backward gradients remains stable and continues to converge.

| Loss | step 0 | step 100 | step 200 | step 300 | step 400 | step 500 | step 600 |
|---|---|---|---|---|---|---|---|
| TAH-QUANT-4bit | 2.87 | 2.71 | 2.63 | 2.59 | 2.56 | 2.55 | 2.52 |
| NAIVE-4bit | 2.87 | 3.04 | 3.16 | 3.21 | 3.49 | 4.53 | 6.04 |

Table 11: Training loss with 4-bit backward gradient quantization.

**Numerical stability in long-horizon training.** During long-horizon pretraining, we observed a single instability event where the loss became `NaN` around 1.9B tokens when using an 8-bit fixed-point compressor for backward gradients. Our investigation indicates that the issue is not caused by the forward TAH-QUANT design (activation quantization), but by insufficient numerical precision in the backward fixed-point path. Increasing the backward precision from 8-bit to 10-bit and resuming from the checkpoint eliminated the issue and restored stable training. This motivates studying more robust backward quantization under aggressive bit budgets; as shown in Table 11, TAH-QUANT remains stable even at 4-bit, whereas naive fixed-point quantization diverges.

**Gradient scaling.** Gradient magnitudes can be small in practice (e.g., on the order of $10^{-5}$ in our setting), which makes ultra-low-bit quantization particularly sensitive to rounding and underflow. To improve numerical robustness, we rescale gradients before quantization and invert the scaling after dequantization:

$$\hat{\mathbf{g}} = Q(c\mathbf{g})/c, \tag{24}$$

where $Q(\cdot)$ denotes the chosen quantizer and $c$ is a constant scaling factor.

# J  Empirical Verification of Assumption 4.4 on the Main Models

In the main text, Assumption 4.4 is verified on `Gemma2-2B`. To address the concern that it should be checked on the models we actually train, we repeat the verification on five models spanning the scales and families we study—the three main-experiment models `GPT-2XL`, `LLaMA-3.2-1B`, and `Qwen2.5-3B`, plus two larger `LLaMA` scale points (`LLaMA-3.2-3B` and `LLaMA-3.1-8B`)—using the same corpus (`WikiText`) for all five so the numbers are directly comparable, at the deployment pipeline depth PP=4 and the default configuration (tile size 64, 80% `INT4`, Hadamard threshold $\tau$=2), measured at batch size 64. We report the squared relative errors of both inequalities of Assumption 4.4: the per-batch error $\|\hat{\mathbf{g}} - \mathbf{g}\|^2/\|\mathbf{g}\|^2$ of (9) (averaged over batches) and the expectation error $\|\mathbb{E}\hat{\mathbf{g}} - \mathbb{E}\mathbf{g}\|^2/\|\mathbb{E}\mathbf{g}\|^2$ of (10) (a single value comparing the dataset-averaged gradients). Table 12 reports these; all quantities are far below 1, so Assumption 4.4 holds on every model tested—the assumption only requires a relative error below 1 (i.e. some $\delta \in (0,1]$).

Table 12: Empirical verification of Assumption 4.4 on these five models (`WikiText`, PP=4, default config $G$=64/80% `INT4`/$\tau$=2, batch size 64). Entries are squared relative gradient errors (the first inequality averaged over batches); all lie well below 1.

| Model | (9): $\|\hat{\mathbf{g}} - \mathbf{g}\|^2/\|\mathbf{g}\|^2$ | (10): $\|\mathbb{E}\hat{\mathbf{g}} - \mathbb{E}\mathbf{g}\|^2/\|\mathbb{E}\mathbf{g}\|^2$ |
|---|---|---|
| `GPT-2XL` | 0.005 | 0.001 |
| `LLaMA-3.2-1B` | 0.154 | 0.176 |
| `LLaMA-3.2-3B` | 0.035 | 0.013 |
| `LLaMA-3.1-8B` | 0.047 | 0.015 |
| `Qwen2.5-3B` | 0.147 | 0.109 |

**Effect of the design parameters.** The relative error in Assumption 4.4 is not a fixed universal constant; it is controlled by the quantizer's design parameters—exactly the knobs raised in the review. Table 13 sweeps them on `LLaMA-3.2-1B` and `LLaMA-3.1-8B` (same `WikiText`/PP=4/bs=64 protocol). Three effects, all consistent with the analysis in Appendix M: (i) a smaller tile size $G$ lowers the error (each scale is then shared across fewer, more homogeneous channels); (ii) a higher `INT4` ratio lowers it further, trading communication bits for fidelity (the accuracy side of this trade-off is swept in Appendix H); and (iii) the Hadamard transform is essential—removing it produces the largest errors we observe and, on `LLaMA-1B`, is the only setting that pushes the per-batch error above 1. Moreover, within the `LLaMA` family the expectation error does *not* grow with model size: it *shrinks* from 0.176 at `1B` to 0.013 and 0.015 at `3B` and `8B`, evidence that the coefficient is not driven by the vector dimension. The assumption is therefore not invoked in a vacuum: it is an empirically measurable, design-controllable quantity that our fine-grained tile-wise, adaptive, Hadamard scheme keeps comfortably within the required range—which is precisely why low-bit (and necessarily *biased*) activation quantization remains stable here.

Table 13: Design-parameter ablation of the Assumption 4.4 error (squared per-batch error (9), averaged over batches; `WikiText`, PP=4, bs=64). Smaller tiles and a higher `INT4` ratio each lower the error; removing the Hadamard transform is by far the worst and breaks the bound on `LLaMA-1B`.

| Setting | LLaMA-1B | LLaMA-8B |
|---|---|---|
| default ($G$=64, 80% `INT4`, $\tau$=2) | 0.154 | 0.047 |
| tile size $G$=32 | 0.065 | 0.035 |
| `INT4` ratio 50% | 0.195 | 0.078 |
| Hadamard off ($\tau$=$\infty$) | 1.133 | 0.316 |

# K   Pretraining: Quantized Baselines and Downstream Evaluation

To complement the training-loss and validation-perplexity curves of the from-scratch `LLaMA-3.2-1B` run (Table 1), we evaluate a full set of compression arms at a fixed token budget ($\sim$1 B tokens) on both held-out perplexity and a downstream task. All arms share the identical training configuration and differ only in the forward activation compressor: `uncompressed` (BF16), `tile-INT4` (tile quantization without the Hadamard transform), `TAH-Quant` (ours), `tile-INT3`, and `channel-INT4` (a coarse per-channel quantizer). We report the clean validation perplexity (computed with the compressor disabled) and zero-shot accuracy on **LAMBADA** (`lambda_openai`), a last-word cloze benchmark that probes long-range context modeling. We also ran a short uncompressed `FP32` reference under the identical setup; its training loss matched the `BF16` uncompressed run over the checked early-training interval, so we adopt `BF16` as the uncompressed reference for the full run.

Table 14 shows that the 4-bit arms are at parity with the uncompressed baseline—`TAH-Quant` matches `uncompressed` and `tile-INT4` on both perplexity and LAMBADA accuracy—while the 3-bit and per-channel arms are marginally worse, mirroring the perplexity ordering. This indicates that the activation compression of TAH-QUANT preserves both held-out perplexity and downstream quality.

Table 14: From-scratch `LLaMA-3.2-1B` at ~1 B tokens: clean validation perplexity and zero-shot LAMBADA accuracy across compression arms. Lower perplexity (↓) and higher LAMBADA accuracy (↑) are better.

| Arm | val. PPL ↓ | LAMBADA acc ↑ |
|---|---|---|
| uncompressed (BF16) | 35.60 | 0.109 |
| **TAH-Quant (ours)** | 35.63 | 0.114 |
| tile-INT4 | 35.59 | 0.106 |
| tile-INT3 | 36.42 | 0.101 |
| channel-INT4 | 36.48 | 0.110 |

## L    Forward/Backward Compute–Communication Overlap

TAH-QUANT compresses forward activations aggressively (3–4 bits) but uses a milder fixed-point quantizer (6–8 bits) for backward gradients. This asymmetry follows from the compute/communication structure of pipeline-parallel training: within a stage, the backward pass performs roughly twice the computation of the forward pass, so the gradient transmission has a proportionally larger window in which to overlap communication with computation. Figure 5 depicts this for a single micro-batch, and Figure 6 shows how the sends overlap computation across stages under a two-stage pipeline schedule. Both are conceptual schematics; exact durations are hardware- and model-dependent.

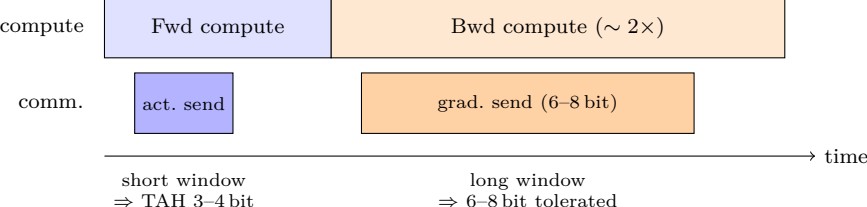

Figure 5: Single-micro-batch schematic. The small, aggressively compressed activation send fits inside the short forward-compute window, while the larger gradient send overlaps the ~2× longer backward-compute window even at a milder 6–8-bit precision.

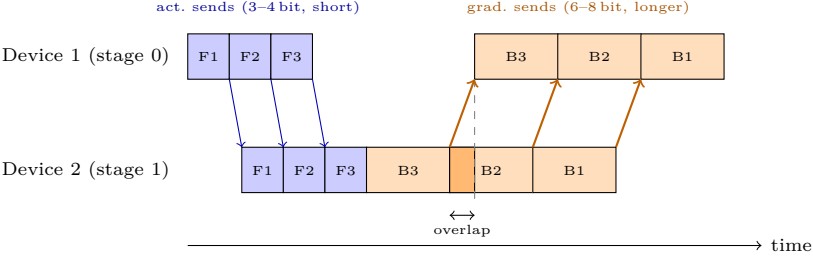

Figure 6: Two-stage pipeline schedule (forward blue; backward orange, drawn ~2× wider to reflect the longer backward compute). Slanted arrows are the inter-stage sends, whose horizontal extent is the transmission time: the TAH-QUANT-compressed activations (blue, stage 0→1, 3–4 bit) are *shorter* than the larger 6–8-bit gradients (orange, stage 1→0), because activations are compressed more aggressively. Each send is transmitted while the sending stage computes its next micro-batch, so communication overlaps computation; the shaded window highlights one such overlap.

## M    On the Derivability of Assumption 4.4

Assumption 4.4 is stated as a hypothesis on the quantized *weight gradient* rather than derived from the quantizer. This appendix separates what the TAH-QUANT design derives at the *activation* level from what

must remain an assumption at the *gradient* level, and shows that the residual assumption is unavoidable for a nonlinear objective and standard in the biased-compression literature. These results justify Assumption 4.4 rather than modify it, and leave Theorem 4.5 unchanged; the empirical validation of Assumption 4.4 on the main models is given separately in Appendix J. Throughout, $g_t = \nabla F(x_t; \xi_t)$ is the stochastic weight gradient and $\hat{g}_t$ the gradient produced after the compressed forward/backward pipeline, as in Assumption 4.4; unsubscripted norms are Euclidean.

## M.1 Assumption 4.4 is a contractive (biased) compressor condition

Inequality (9) has the same relative-contraction form as the biased-compressor condition $\mathbb{E}_C \|C(v) - v\|^2 \leq (1 - \delta) \|v\|^2$ used for Top-$k$, low-rank, and error-feedback compressors (Stich et al., 2018; Karimireddy et al., 2019; Beznosikov et al., 2020; Richtárik et al., 2021). Two points clarify its status. First, the coefficient in question is the $(1 - \delta)$ of Assumption 4.4 itself, a relative error of the weight gradient $\hat{g}_t$, and it is a scale-invariant contraction factor in $(0, 1]$; by contrast, unbiased compressors such as QSGD are characterized by a variance parameter $\omega$ in $\mathbb{E}_C \|C(v) - v\|^2 \leq \omega \|v\|^2$, where $\omega$ may depend on the dimension and the number of quantization levels (Alistarh et al., 2016)—a different object. Second, Assumption 4.4 does not assume $\mathbb{E}[\hat{g}_t] = g_t$, which is why the analysis carries the *separate* bias control (10); under unbiasedness (10) would be automatic. The analysis is therefore of the biased/contractive type rather than the unbiased-compression template, and is strictly more delicate (a naive biased compressor without error feedback can diverge (Karimireddy et al., 2019)). Whether the coefficient can be connected to the TAH-QUANT design is established in the remainder of this appendix: the design parameters control the activation- and backward-level relative errors (Sections M.2–M.3), which propagate into the gradient coefficient (Section M.4). Table 15 places Assumption 4.4 in the compression taxonomy.

| | Unbiased compression | Assumption 4.4 (ours) |
|---|---|---|
| Defining condition | $\mathbb{E}[C(v)] = v, \quad \mathbb{E} \|C(v) - v\|^2 \leq \omega \|v\|^2$ | $\|\hat{g}_t - g_t\|^2 \leq (1 - \delta) \|g_t\|^2$ (contraction) |
| Bias handled by | automatically ($\mathbb{E}[C(v)] = v$) | a separate hypothesis, (10) |
| Representative work | QSGD (Alistarh et al., 2016) | Top-$k$/EF (Stich et al., 2018; Karimireddy et al., 2019; Richtárik et al., 2021) |

Table 15: Assumption 4.4 belongs to the biased/contractive-compressor family, not the unbiased family; the proof reflects this through the separate bias condition (10).

## M.2 The activation-level error is derivable

TAH-QUANT compresses each boundary-activation tile $\alpha \in \mathbb{R}^G$ by an optional Hadamard/pivot rotation followed by $b$-bit asymmetric uniform quantization $Q_b$ with a per-tile scale (Algorithm 1). Write $\mu(\alpha) := \|\alpha\|_\infty / \|\alpha\|_2 \in [1/\sqrt{G}, 1]$ for the tile incoherence, the quantity through which the dynamic range enters.

**Lemma M.1** (Tile-wise quantization error)**.** *Let $Q_b$ be deterministic round-to-nearest $b$-bit asymmetric uniform quantization with the scale set from the exact tile range (no clipping or saturation). For any $\alpha \neq 0$,*

$$\frac{\|Q_b(\alpha) - \alpha\|^2}{\|\alpha\|^2} \leq \frac{G \, \mu(\alpha)^2}{(2^b - 1)^2}. \tag{25}$$

*Proof.* The step is $\Delta = R/(2^b - 1)$ with range $R = \max_i \alpha_i - \min_i \alpha_i \leq 2 \|\alpha\|_\infty$, and each coordinate satisfies $|Q_b(\alpha)_i - \alpha_i| \leq \Delta/2$. Hence $\|Q_b(\alpha) - \alpha\|^2 \leq G(\Delta/2)^2 = GR^2/(4(2^b - 1)^2) \leq G \|\alpha\|_\infty^2 / (2^b - 1)^2$; dividing by $\|\alpha\|^2$ gives (25). □

The design enters through $b$, quadratically, and through $\mu(\alpha)$; the Hadamard rotation exists to make $\mu$ small.

**Lemma M.2** (Randomized-Hadamard incoherence). *Let $H_G \in \{\pm 1\}^{G \times G}$ be a Hadamard matrix ($H_G H_G^\top = GI$, so $G$ is a Hadamard order, e.g. a power of two, as in our tiling) and $\widetilde{H} := H_G D / \sqrt{G}$ with $D = \mathrm{Diag}(\varepsilon_1, \dots, \varepsilon_G)$, $\varepsilon_i$ i.i.d. Rademacher. For any fixed $\alpha \neq 0$, with probability at least $1 - \beta$,*

$$\mu(\widetilde{H}\alpha) \leq \sqrt{2 \log(2G/\beta)/G}. \tag{26}$$

*Proof.* $\widetilde{H}$ is orthogonal, so $\left\| \widetilde{H}\alpha \right\|_2 = \|\alpha\|_2$. Each coordinate $[\widetilde{H}\alpha]_k = \sum_i (h_{ki}/\sqrt{G})\varepsilon_i\alpha_i$ is a sum of independent mean-zero terms bounded by $|\alpha_i|/\sqrt{G}$, so Hoeffding's inequality gives $\Pr(|[\widetilde{H}\alpha]_k| \geq s) \leq 2\exp(-Gs^2/(2\|\alpha\|_2^2))$. Taking $s = \|\alpha\|_2 \sqrt{2\log(2G/\beta)/G}$ makes each tail at most $\beta/G$; a union bound over the $G$ coordinates and division by $\left\| \widetilde{H}\alpha \right\|_2 = \|\alpha\|_2$ give (26). $\square$

**Corollary M.3** (Dynamic-range-free tile bound). *With a randomized Hadamard rotation followed by $b$-bit quantization, with probability at least $1 - \beta$,*

$$\frac{\left\| Q_b(\widetilde{H}\alpha) - \widetilde{H}\alpha \right\|^2}{\|\alpha\|^2} \leq \frac{2\log(2G/\beta)}{(2^b - 1)^2}. \tag{27}$$

*The dimension $G$ has cancelled: the bound depends on tile size only logarithmically, and not on the raw dynamic range.*

*Proof.* Apply Lemma M.1 to $\widetilde{H}\alpha$ (using $\left\| \widetilde{H}\alpha \right\|_2 = \|\alpha\|_2$) and insert (26), so $G\mu(\widetilde{H}\alpha)^2 \leq 2\log(2G/\beta)$ on the event of probability at least $1 - \beta$. $\square$

A boundary activation is a concatenation of tiles, so we aggregate.

**Lemma M.4** (Whole-activation error). *Let the activation be tiled as $\alpha_1, \dots, \alpha_N$ of common size $G$ (zero-energy tiles excluded by convention), with per-tile squared relative errors $\rho_j$ and energy weights $w_j = \|\alpha_j\|^2 / \sum_\ell \|\alpha_\ell\|^2$. The whole-activation squared relative error is the energy-weighted average $\sum_j w_j \rho_j$. If every tile uses a randomized Hadamard rotation and (27) is required for all $N$ tiles simultaneously, then allocating $\beta_j = \beta/N$ and a union bound give, with probability at least $1 - \beta$,*

$$\rho_{\mathrm{act}} := \frac{\sum_j \left\| Q_{b_j}(\alpha_j) - \alpha_j \right\|^2}{\sum_j \|\alpha_j\|^2} \leq 2\log\!\left(\frac{2GN}{\beta}\right) \sum_{j=1}^N \frac{w_j}{(2^{b_j} - 1)^2}. \tag{28}$$

*Proof.* The energy-weighted identity is the ratio with numerator and denominator split tile-wise. The union bound over the $N$ events of Corollary M.3, each of failure probability at most $\beta/N$, makes all per-tile bounds hold together with probability at least $1 - \beta$; substituting gives (28). $\square$

The adaptive-bit rule enters (28) through the bit assignment: with INT4/INT3 and INT4 energy share $p_E := \sum_{j:b_j=4} w_j$, $\rho_{\mathrm{act}} \leq 2\log(2GN/\beta)\,[p_E/15^2 + (1 - p_E)/7^2]$. The top-$p$ selection is a tile-count rule, so $p_E$ equals the count fraction only when the selected tiles carry a proportional share of the energy; the entropy criterion tends to make $p_E$ at least the count fraction, an empirical regularity we do not rely on theoretically. Table 16 summarizes how each design parameter enters $\delta_{\mathrm{act}} = 1 - \rho$.

At $G = 64$ and per-tile $\beta = 0.05$, $2\log(2G/\beta) = 2\log(2560) \approx 15.7$, so the single-tile bounds are $\rho \lesssim 15.7/15^2 \approx 0.070$ for INT4 ($\delta_{\mathrm{act}} \gtrsim 0.93$) and $\rho \lesssim 15.7/7^2 \approx 0.32$ for INT3 ($\delta_{\mathrm{act}} \gtrsim 0.68$); an INT4 energy share $p_E \approx 0.8$ gives $\rho_{\mathrm{act}} \lesssim 0.12$ as a per-tile illustration (the simultaneous whole-activation guarantee replaces 15.7 by $2\log(2GN/\beta)$, e.g. $\approx 0.19$ at $N = 128$, still logarithmic in $N$). These activation-level coefficients are numerically compatible with the end-to-end gradient errors of Appendix J, though the two are on different objects.

| Design knob | Enters $\rho$ as | Effect on $\delta_{\text{act}}$ |
|---|---|---|
| bit-width $b$ | $\rho \propto (2^b - 1)^{-2}$ | increases quadratically in $b$ |
| tile size $G$ (rand. Hadamard) | $\rho \leq 2\log(2G/\beta)/(2^b - 1)^2$ | decreases only log-slowly in $G$ |
| high-precision ratio $p_E$ | energy-weighted INT4/INT3 mix (28) | increases with the INT4 energy share |
| outlier threshold $\tau$ | selects tiles rotated to reduce $\mu$ | shrinks $\mu$ on rotated tiles |

Table 16: How the TAH-QUANT design parameters enter the activation-level contraction coefficient $\delta_{\text{act}}$.

**Remark (theorem variant vs. implemented transform).** The dynamic-range-free bound (27) is a property of the randomized-sign Hadamard $\widetilde{H} = H_G D/\sqrt{G}$ (as in QuaRot/SpinQuant (Ashkboos et al., 2024; Liu et al., 2024b)); it is not claimed verbatim for the deterministic pivot-swap of Algorithm 1, which attains $\mu = 1/\sqrt{G}$ for a single dominant outlier but not the high-probability incoherence (26) for arbitrary tiles. Two readings are consistent with the deployed method: instantiate the transform with a shared-seed Rademacher diagonal $D$, at zero communication overhead, making (27) rigorous; or keep the deterministic transform and read the bound conditionally on the observed post-transform incoherence, which Appendix J shows is small in practice. Under the first reading, the probability in (27) is taken over the shared random signs; since Assumption 4.4 takes expectation over the data draw and not over compressor randomness, for a finite evaluated trajectory one may fix a shared seed and condition on the event that the corresponding incoherence bounds hold (this does not assert a uniform guarantee over all possible activations). Either reading leaves the coefficient explicit and neither changes Theorem 4.5.

**Remark (why the deployed transform is conditional and deterministic).** The transform is applied only when the outlier ratio $r = |\alpha_{(1)}|/|\alpha_{(2)}|$ exceeds the threshold $\tau$, and this conditioning is deliberate. A fixed Hadamard matrix maps a low-outlier tile to a concentrated one: a near-constant tile $\alpha \approx c\mathbf{1}$ is sent to $H_G(c\mathbf{1}) = (Gc, 0, \ldots, 0)$, raising $\mu$ from its floor $1/\sqrt{G}$ to 1, so deterministically rotating an already-incoherent tile would *create* an outlier, and the $\tau$-trigger skips exactly those tiles. The deterministic pivot-swap places the realized dominant coordinate where the Hadamard flattens it, attaining $\mu = 1/\sqrt{G}$ for the single-outlier case that dominates activation tiles, with no random seed to synchronize across stages. The randomized bound is a worst-case certificate; the deployed transform is optimized for the benign, single-outlier structure real activations exhibit, and the residual worst-case gap—multi-outlier tiles that escape the $\tau$-trigger—is empirically rare (Appendix J).

### M.3 The backward compressor

The backward path quantizes the activation gradient with a $b_b$-bit fixed-point compressor, $b_b \in \{6, 8\}$, following AQ-SGD (Wang et al., 2022).

**Lemma M.5** (Backward tile error). *For a gradient tile $v \in \mathbb{R}^{G_b}$ under per-tile-scaled $b_b$-bit uniform quantization (dynamic scale, round-to-nearest, no clipping), $\|Q(v) - v\|^2 / \|v\|^2 \leq G_b \mu(v)^2/(2^{b_b} - 1)^2 \leq G_b/(2^{b_b} - 1)^2$. At $G_b = 64$ this is at most $1.6 \times 10^{-2}$ for $b_b = 6$ and $9.8 \times 10^{-4}$ for $b_b = 8$ (norm errors $\approx 0.127$ and $\approx 0.031$, respectively), so 6–8 backward bits are far more forgiving than 3–4 forward bits.*

**Proposition M.6** (Linear pass-through). *For a local linear map $h = Wz$ the weight gradient is $g_W = vz^\top$. If only the backward signal is compressed, $\hat{v} = v + e_v$ with $z$ unchanged, then $\hat{g}_W - g_W = e_v z^\top$ and $\|\hat{g}_W - g_W\|_F / \|g_W\|_F = \|e_v\| / \|v\|$.*

*Proof.* With $z$ fixed, $g_W = vz^\top$ is linear in $v$, so $\hat{g}_W - g_W = e_v z^\top$. Both are rank-one, so $\|e_v z^\top\|_F = \|e_v\| \|z\|$ and $\|vz^\top\|_F = \|v\| \|z\|$, giving the ratio. $\square$

This is a local algebraic identity ($z$ fixed, one linear layer), not a global propagation result: the backward compressor, taken alone, transmits its relative error to the weight gradient without loss. The genuine subtlety is that $v$ is evaluated on the *quantized* forward activation, treated next.

### M.4 From the activation error to the weight gradient

Consider a two-stage pipeline $F(x_A, x_B; \xi) = \phi_B(f_A(x_A; \xi), x_B; \xi)$ with $a = f_A(x_A; \xi)$, $v(a) = \nabla_a \phi_B(a, x_B; \xi)$, and upstream weight gradient $g_A = J_A^\top v(a)$, $J_A = \partial a / \partial x_A$. Forward quantization gives $\hat{a} = a + e_a$, $\|e_a\| \leq \varepsilon_a \|a\|$; the backward signal is $\hat{v} = v(\hat{a}) + e_v$, $\|e_v\| \leq \varepsilon_b \|v(\hat{a})\|$; and $\hat{g}_A = J_A^\top \hat{v}$. Boundary-only quantization leaves stage $A$'s internal forward pass, hence $J_A$, unquantized, so $\hat{g}_A$ uses the same $J_A$ as $g_A$; this is relaxed below.

**Proposition M.7** (Activation-to-gradient perturbation). *If $\|J_A\|_{op} \leq M_A$ and $v(\cdot)$ is $L_v$-Lipschitz in $a$, then*

$$\|\hat{g}_A - g_A\| \leq M_A \big[ \varepsilon_b \|v(a)\| + (1 + \varepsilon_b) L_v \varepsilon_a \|a\| \big]. \tag{29}$$

*Proof.* $\hat{g}_A - g_A = J_A^\top(\hat{v} - v(a))$ with $\hat{v} - v(a) = e_v + [v(\hat{a}) - v(a)]$. By the compressor bound and Lipschitzness, $\|e_v\| \leq \varepsilon_b \|v(\hat{a})\| \leq \varepsilon_b(\|v(a)\| + L_v \|e_a\|)$ and $\|v(\hat{a}) - v(a)\| \leq L_v \|e_a\| \leq L_v \varepsilon_a \|a\|$; combining and using $\|J_A^\top(\cdot)\| \leq M_A \|\cdot\|$ gives (29). $\square$

The bound (29) is *additive*: its right-hand side scales with $\|v(a)\|$ and $\|a\|$, not with $\|g_A\|$, and $\|g_A\| \leq M_A \|v(a)\|$ is only an upper bound on $\|g_A\|$.

**Remark (relaxing the unchanged-Jacobian assumption).** The same-$J_A$ setup is not needed. If internal quantization perturbs the Jacobian to $\hat{J}_A$, then $\hat{g}_A - g_A = \hat{J}_A^\top(\hat{v} - v(a)) + (\hat{J}_A - J_A)^\top v(a)$; if $\left\|\hat{J}_A\right\|_{op} \leq M_A$ and $\left\|\hat{J}_A - J_A\right\|_{op} \leq \varepsilon_J$, then (29) gains a single additive term $\varepsilon_J \|v(a)\|$, and the coefficient below becomes $\kappa_A = (M_A \varepsilon_b + \varepsilon_J)/\gamma_A + M_A(1 + \varepsilon_b) L_v \Gamma_A \varepsilon_a$; the boundary-only case is $\varepsilon_J = 0$. The operator-norm bound is on the realized $\hat{J}_A$, while the non-degeneracy margin of Assumption P below remains on the true $J_A$ defining $g_A$; a margin on $\hat{J}_A$, if wanted, follows as $\left\|\hat{J}_A^\top z\right\| \geq (\gamma_A - \varepsilon_J) \|z\|$ whenever $\varepsilon_J < \gamma_A$, though the derivation does not need it.

The downstream parameters $x_B$ act directly on the quantized activation, giving a simpler companion bound.

**Proposition M.8** (Downstream-gradient perturbation). *If $g_B = \nabla_{x_B} \phi_B(a, x_B; \xi)$ and $\nabla_{x_B} \phi_B(\cdot, x_B; \xi)$ is $L_B$-Lipschitz in its activation argument, then $\|\hat{g}_B - g_B\| \leq L_B \varepsilon_a \|a\|$.*

*Proof.* Immediate from $L_B$-Lipschitzness and $\|\hat{a} - a\| \leq \varepsilon_a \|a\|$. $\square$

Stacking the stages over the disjoint parameter blocks $x_A, x_B$, $\|\hat{g} - g\|^2 = \|\hat{g}_A - g_A\|^2 + \|\hat{g}_B - g_B\|^2$, so Propositions M.7–M.8 assemble into an additive whole-gradient bound (one $(J, L_v, L_B)$ triple per boundary in a multi-stage pipeline). This is the reach of the derivation without further hypotheses; whether it becomes the relative form (9) is decided by the following.

**Proposition M.9** (No uniform relative bound from activation fidelity alone). *There is no constant $\delta > 0$ depending only on the activation fidelity such that $\|\hat{g} - g\|^2 \leq (1 - \delta) \|g\|^2$ holds at every parameter point, even for arbitrarily small $|\hat{a} - a| \leq \varepsilon |a|$.*

*Proof.* Take any $c \neq 0$, let $a = x$ and $F(x) = \frac{1}{2}(a - c)^2$, so $g = x - c$ and $\hat{g} = \hat{a} - c = g + e$ under $\hat{a} = a + e$. At $x = c$, $g = 0$ while $\hat{g} - g = e \neq 0$ whenever $e \neq 0$, so $\|\hat{g} - g\|^2 > 0 = (1 - \delta) \|g\|^2$ for every $\delta$; as $x \to c$ the ratio $|e|/|g|$ is unbounded. $\square$

Proposition M.9 rules out a uniform, sample-wise relative bound derived from activation fidelity alone; it does not contradict Assumption 4.4, which is empirically observed to hold along the evaluated trajectory (Appendix J). This is why every contraction-based compression analysis assumes the contraction on the iterate rather than deriving it from the encoder (Stich et al., 2018; Karimireddy et al., 2019; Beznosikov et al., 2020; Richtárik et al., 2021). The exact relative form follows from one further, trajectory-local condition, stated explicitly rather than folded silently into Assumption 4.4; it is a strong non-degeneracy condition, not a standard fact.

**Assumption P** (Local gradient non-degeneracy). *Let $\mathcal{V}_A = \operatorname{span}\{v(a), v(\hat{a}), e_v\}$ be spanned by the backward signals arising at the boundary. There are constants $\gamma_A, \Gamma_A, \Gamma_B > 0$ with $\left\lVert J_A^\top z \right\rVert \geq \gamma_A \left\lVert z \right\rVert$ for all $z \in \mathcal{V}_A$, and $\lVert a \rVert \leq \Gamma_A \lVert g_A \rVert$, $\lVert a \rVert \leq \Gamma_B \lVert g_B \rVert$.*

**Proposition M.10** (Sufficient condition for Assumption 4.4). *Under Propositions M.7–M.8 and Assumption P, $\lVert \hat{g}_A - g_A \rVert \leq \kappa_A \lVert g_A \rVert$ with $\kappa_A = \frac{M_A}{\gamma_A}\varepsilon_b + M_A(1 + \varepsilon_b)L_v\Gamma_A\varepsilon_a$, and $\lVert \hat{g}_B - g_B \rVert \leq L_B\Gamma_B\varepsilon_a \lVert g_B \rVert$. Hence with $\kappa = \max\{\kappa_A, L_B\Gamma_B\varepsilon_a\}$, the assembled gradient satisfies $\lVert \hat{g} - g \rVert^2 \leq \kappa^2 \lVert g \rVert^2$, so if $\kappa < 1$ then (9) holds with $\delta = 1 - \kappa^2$.*

*Proof.* Divide (29) by $\lVert g_A \rVert$ and use $\lVert v(a) \rVert \leq \gamma_A^{-1} \left\lVert J_A^\top v(a) \right\rVert = \gamma_A^{-1} \lVert g_A \rVert$ (valid since $v(a) \in \mathcal{V}_A$) and $\lVert a \rVert \leq \Gamma_A \lVert g_A \rVert$; divide Proposition M.8 by $\lVert g_B \rVert$ and use $\lVert a \rVert \leq \Gamma_B \lVert g_B \rVert$; square and add. $\square$

### M.5 The bias condition (10) and worked examples

Condition (10) is stronger than (9): even granting (9) sample-wise, Jensen's inequality gives only $\lVert \mathbb{E}_\xi[\hat{g}_t] - \nabla f(x_t) \rVert^2 \leq (1 - \delta)\mathbb{E}_\xi \lVert g_t \rVert^2 \leq (1 - \delta)(\lVert \nabla f(x_t) \rVert^2 + \sigma^2)$ by Assumption 4.3, an extra $(1 - \delta)\sigma^2$. Moreover, since TAH-QUANT compresses activations and the activation-to-gradient map is nonlinear, an unbiased activation rounding does not make $\hat{g}_t$ unbiased: with a twice-differentiable map of bounded second derivative, a Taylor expansion leaves a second-order term $O(\mathbb{E} \lVert e_a \rVert^2)$. We therefore treat (10) as a separate compressor-side bias condition, validated in Appendix J and, on Gemma2-2B, in the main text (below 0.1).

**Two worked examples.** *(i)* $G = 4$. With $\frac{1}{2}H_4$, a single-outlier tile $\alpha = (a, 0, 0, 0)$ has $\mu = 1$; after rotation $\frac{1}{2}H_4\alpha^\top = (\frac{a}{2}, \frac{a}{2}, \frac{a}{2}, \frac{a}{2})$, so $\mu = 1/2 = 1/\sqrt{G}$ and $G\mu^2$ drops from 4 to 1. A flat tile $(c, c, c, c)$ has $\mu = 1/2$; rotating it gives $(2c, 0, 0, 0)$, so $\mu = 1$ and $G\mu^2$ rises to 4—which is why the $\tau$-trigger rotates the spike ($r = \infty$) and skips the flat tile ($r = 1$). *(ii) Jacobian cancellation.* With $J_A^\top = [1, -1]$ and $v = (1, 1 + t)$, $g_A = -t$; perturbing only the backward signal by $e_v = (\eta, 0)$ gives a backward relative error $\approx \eta/\sqrt{2}$ but $\hat{g}_A - g_A = \eta$, so the gradient relative error $\eta/t \to \infty$ as $t \to 0$, since $\left\lVert J_A^\top v \right\rVert / \lVert v \rVert = t/\sqrt{2} \to 0$ violates Assumption P's margin. These illustrate, respectively, the conditional rotation and the necessity of the non-degeneracy margin.

**Scope.** At the quantizer level, Lemma M.1, Corollary M.3, and Lemma M.4 derive the activation-level relative error explicitly in $b$, $G$, and the high-precision ratio, with only logarithmic dependence on dimension under randomized rotation; the backward compressor transmits its error cleanly (Lemma M.5, Proposition M.6). At the propagation level, Propositions M.7–M.8 give an additive weight-gradient bound unconditionally, and Proposition M.10 the relative form of Assumption 4.4 under the single non-degeneracy margin of Assumption P; Proposition M.9 shows that a uniform sample-wise derivation from activation fidelity alone is impossible for a nonlinear objective, so stating it as an assumption at the gradient level is the standard modelling choice. Relative to the biased-compression works, which posit the contraction on the compressed vector, TAH-QUANT additionally derives the activation- and backward-level terms that feed the gradient coefficient. Theorem 4.5 is unchanged, and Assumption 4.4 is validated empirically on the main models in Appendix J.

## N  Extension of the Convergence Analysis to Adam

The main-text guarantee (Theorem 4.5) is stated for momentum SGD. Here we give the complete, self-contained generalization to the Adam optimizer used in our pretraining experiments, including the extra assumption it requires, a proof that some such assumption is *necessary*, and the full convergence proof. The result **reduces exactly to Theorem 4.5 when the adaptive preconditioner is the identity**, so it is a strict generalization. Throughout we keep the manuscript's convention in which $\beta_1 \in (0, 1)$ multiplies the *current* gradient in the first-moment recursion, so the small-$\beta_1$ regime of our theorems corresponds to the standard heavy-momentum Adam regime $\beta_1^{\text{Adam}} = 1 - \beta_1 \to 1$.

### N.1 Problem setup and the Adam iteration

We study the same stochastic problem as in the main text, $\min_{x\in\mathbb{R}^d} f(x) := \mathbb{E}_{\xi\sim\mathcal{D}}[F(x;\xi)]$, with stochastic gradient $g_t := \nabla F(x_t;\xi_t)$ and quantized stochastic gradient $\hat{g}_t$ from the TAH-QUANT operator. The only change is that the momentum-SGD update is replaced by the Adam iteration, in its (bias-correction-free) core form:

$$m_t = (1-\beta_1)\, m_{t-1} + \beta_1\, \hat{g}_t, \tag{30}$$

$$v_t = \beta_2\, v_{t-1} + (1-\beta_2)\, \hat{g}_t^{\odot 2}, \tag{31}$$

$$x_{t+1} = x_t - \eta\, D_t\, m_t, \qquad D_t := \left(\mathrm{Diag}(v_t)^{1/2} + \varepsilon I\right)^{-1}, \tag{32}$$

where $\hat{g}_t^{\odot 2}$ is the coordinatewise square, $\varepsilon > 0$ the numerical floor, and $D_t$ the diagonal symmetric positive-definite adaptive preconditioner. Setting $D_t \equiv I$ recovers momentum SGD. We write $e_t := m_t - \nabla f(x_t)$ for the first-moment tracking error and, for symmetric positive-definite $A$, $\|u\|_A^2 := u^\top A\, u$; unsubscripted norms are Euclidean.

**Remark (bias correction).** Textbook Adam rescales the moments by deterministic bias-correction factors that decrease monotonically to 1; the second-moment factor merely rescales the spectral constants $\mu, M$ of Assumption A5 by bounded factors, and the first-moment factor is confined to a finite, $T$-independent warm-up that decays geometrically, hence does not affect the $O(1/\sqrt{T})$ rate. We therefore analyze the constant-step core form, as is standard in non-asymptotic Adam analyses (Défossez et al., 2022).

### N.2 Assumptions

The four assumptions of the main text (Assumptions 4.1–4.4) are kept verbatim; we recall them with $\delta \in (0,1]$ the quantization-fidelity coefficient of Assumption 4.4.

**(A4.1) (Lower boundedness)** $\inf_{x\in\mathbb{R}^d} f(x) > -\infty$.

**(A4.2) (L-smoothness)** $\|\nabla f(x) - \nabla f(y)\| \leq L\,\|x-y\|$ for all $x,y$.

**(A4.3) (Stochastic gradient)** $\mathbb{E}[\nabla F(x_t;\xi_t)] = \nabla f(x_t)$ and $\mathbb{E}\,\|\nabla F(x_t;\xi_t) - \nabla f(x_t)\|^2 \leq \sigma^2$.

**(A4.4) (Quantization error)** For some $\delta \in (0,1]$, $\|\hat{g}_t - g_t\|^2 \leq (1-\delta)\,\|g_t\|^2$ and $\|\mathbb{E}_{\xi_t}[\hat{g}_t] - \nabla f(x_t)\|^2 \leq (1-\delta)\,\|\nabla f(x_t)\|^2$.

The momentum-SGD proof closes in Euclidean geometry because the step is the scalar multiple $-\eta m_t$. Adam replaces this with $-\eta D_t m_t$, a *variable-metric* step. The single new ingredient is a uniform control on that metric.

**Assumption A5** (Bounded adaptive preconditioner). *There exist constants $0 < \mu \leq M < \infty$ such that, almost surely and for all $t \geq 0$, $\mu I \preceq D_t \preceq M I$. We write $\kappa_D := M/\mu \geq 1$ for the condition number of the preconditioner family.*

**Discussion of Assumption A5.** (i) For momentum SGD ($D_t \equiv I$) it holds with $\mu = M = 1$, $\kappa_D = 1$; A5 is vacuous in the regime of Theorem 4.5, which our result contains as the case $\kappa_D = 1$. (ii) A5 is implied by a coordinatewise gradient bound, the standard hypothesis for Adam analyses (Kingma & Ba, 2015; Reddi et al., 2018; Chen et al., 2019; Défossez et al., 2022): if $\|\hat{g}_t\|_\infty \leq G_\infty$, then $0 \leq v_{t,i} \leq G_\infty^2$, so each diagonal entry of $D_t$ lies in $[\frac{1}{G_\infty+\varepsilon}, \frac{1}{\varepsilon}]$ and A5 holds with

$$\mu = \frac{1}{G_\infty+\varepsilon}, \qquad M = \frac{1}{\varepsilon}, \qquad \kappa_D = \frac{G_\infty+\varepsilon}{\varepsilon}. \tag{33}$$

In finite-precision implementations this coordinatewise bound is empirically benign; we nonetheless keep it explicit as an assumption rather than claiming it by construction.

### N.3 An Adam-specific condition is unavoidable

**Proposition N.1** (Insufficiency of A4.1–A4.4 for vanilla Adam)**.** *Assumptions 4.1–4.4 alone do not imply convergence of the vanilla Adam iteration* (30)–(32)*. Hence the momentum-SGD argument cannot be transported to Adam without an additional condition controlling $D_t$; Assumption A5 is one such proof-compatible condition.*

*Justification. Analytic obstruction.* The momentum-SGD descent step is $x_{t+1} - x_t = -\eta m_t$, which under $L$-smoothness produces the clean Euclidean term $\frac{\eta}{2} \|\nabla f(x_t)\|^2$. For Adam the step $-\eta D_t m_t$ produces instead the variable-metric quantity $\langle \nabla f(x_t), D_t \nabla f(x_t) \rangle$. Assumptions 4.1–4.4 place no lower bound on the smallest eigenvalue of $D_t$, which can be arbitrarily close to 0; then $\langle \nabla f(x_t), D_t \nabla f(x_t) \rangle$ carries no usable fraction of $\|\nabla f(x_t)\|^2$ and the descent argument cannot close. The uncontrolled quantity is exactly the spectrum of $D_t$, which A5 supplies. *Algorithmic obstruction.* Reddi et al. (2018) exhibit convex problems with bounded gradients on which vanilla Adam with constant $(\beta_1, \beta_2)$ provably fails to converge. Specializing to $\hat{g}_t = g_t$ (i.e. $\delta = 1$) yields a problem satisfying A4.1–A4.4 on which Adam diverges; hence no theorem "A4.1–A4.4 $\Rightarrow$ Adam converges" can exist, and some Adam-specific hypothesis is logically necessary. Modern analyses supply it in different forms (long-term memory (Reddi et al., 2018), bounded preconditioner/gradients (Chen et al., 2019; Défossez et al., 2022), conditions on the moment sequences (Zou et al., 2019), or the region $\beta_1 < \sqrt{\beta_2}$ (Zhang et al., 2022)); A5 is the representative compatible with our existing proof, and (33) shows it follows from the bounded-gradient hypothesis used by several of these works. $\qquad\square$

### N.4 Main result

**Theorem N.2** (TAH-QUANT with Adam; generalization of Theorem 4.5)**.** *Suppose Assumptions 4.1–4.4 and A5 hold and $\delta \in (0, 1)$. Assume the* compatibility condition

$$\kappa_D \left(1 - \tfrac{\delta}{2}\right)(1 + 6\beta_1) \ \leq \ 1 - \tfrac{\delta}{4}, \tag{34}$$

*and the* step-size condition $\eta \leq \min\left\{ \frac{1}{2LM}, \ \frac{\beta_1}{LM}\sqrt{\frac{\delta}{8}} \right\}$*. Then the Adam iteration* (30)–(32) *equipped with TAH-QUANT satisfies*

$$\frac{1}{T+1} \sum_{t=0}^{T} \mathbb{E} \|\nabla f(x_t)\|^2 \leq \frac{8\big(f(x_0) - \inf_x f\big)}{\delta\mu\eta(T+1)} + \frac{8M\|m_0 - \nabla f(x_0)\|^2}{\delta\mu\beta_1(T+1)} + \frac{24M\beta_1\sigma^2}{\delta\mu}. \tag{35}$$

**Remark (exact reduction to Theorem 4.5).** With $D_t \equiv I$ we have $\mu = M = 1$, $\kappa_D = 1$. Then (34) reads $(1 - \delta/2)(1 + 6\beta_1) \leq 1 - \delta/4$, equivalent to $\beta_1 \leq \delta/(24 - 12\delta)$; the step-size condition becomes $\eta \leq \min\{1/(2L), (\beta_1/L)\sqrt{\delta/8}\}$; and (35) becomes *verbatim* Theorem 4.5. Thus Theorem N.2 is a strict generalization, exposing the extra price of adaptivity through the factors $M/\mu = \kappa_D$ and the tightened window (34).

**Corollary N.3** ($O(1/\sqrt{T})$ rate)**.** *If, in addition, the preconditioner is strictly well conditioned, $\kappa_D < \frac{4-\delta}{4-2\delta}$, then choosing $\beta_1 = \theta/\sqrt{T+1}$ and $\eta = \frac{\beta_1}{LM}\sqrt{\delta/8}$ (for a constant $\theta > 0$) makes* (34) *and the step-size condition hold for all large $T$, and*

$$\frac{1}{T+1} \sum_{t=0}^{T} \mathbb{E} \|\nabla f(x_t)\|^2 \leq \frac{C}{\sqrt{T+1}} = O\Big(\tfrac{1}{\sqrt{T+1}}\Big), \quad C = \frac{8\sqrt{8}\,LM\Delta_0}{\delta^{3/2}\mu\theta} + \frac{8M\|e_0\|^2}{\delta\mu\theta} + \frac{24M\sigma^2\theta}{\delta\mu}, \tag{36}$$

*with $\Delta_0 := f(x_0) - \inf_x f$, $e_0 := m_0 - \nabla f(x_0)$. This matches the $O(1/\sqrt{T})$ rate of momentum SGD (Corollary 4.6), with the constant inflated only by the conditioning factor $\kappa_D$.*

### N.5 Proofs

The proof keeps the entire algebra of the main-text momentum-contraction lemma (Lemma B.2); the only genuinely new component is a variable-metric replacement for the Euclidean descent lemma (Lemma B.1).

|  | **Momentum SGD** (Thm. 4.5) | **Vanilla Adam**, A4.1– A4.4 only | **Adam** (Thm. N.2) |
|---|---|---|---|
| Update | $x_{t+1} = x_t - \eta m_t$ | $x_{t+1} = x_t - \eta D_t m_t$ | $x_{t+1} = x_t - \eta D_t m_t$ |
| Original proof closes? | yes | **no** (Prop. N.1) | yes, via A5 |
| Extra assumption | none | at least one Adam-specific | $\mu I \preceq D_t \preceq MI +$ (34) |
| Nonconvex stochastic | $O(1/\sqrt{T})$ | fails in general | $O(1/\sqrt{T})$ |

Table 17: Where the momentum-SGD analysis transfers to Adam, and the single structural addition required.

**Lemma N.4** (Variable-metric descent). *Under Assumptions 4.2 and A5, the Adam update* (32) *satisfies, for every $t \geq 0$,*

$$f(x_{t+1}) \leq f(x_t) - \frac{\eta\mu}{2} \|\nabla f(x_t)\|^2 - \left(\frac{1}{2\eta} - \frac{LM}{2}\right) \|x_{t+1} - x_t\|^2_{D_t^{-1}} + \frac{\eta M}{2} \|e_t\|^2. \tag{37}$$

*Proof.* By $L$-smoothness, $f(x_{t+1}) \leq f(x_t) + \langle \nabla f(x_t), x_{t+1} - x_t \rangle + \frac{L}{2} \|x_{t+1} - x_t\|^2$. Write $s_t := x_{t+1} - x_t = -\eta D_t m_t$ and let $\langle u, w \rangle_{D_t} := u^\top D_t w$, $\|u\|^2_{D_t} := u^\top D_t u$. Since $D_t$ is symmetric, $\langle \nabla f(x_t), D_t m_t \rangle = \langle \nabla f(x_t), m_t \rangle_{D_t}$, and the $D_t$-polarization identity with $a = \nabla f(x_t)$, $b = m_t$ (so $a - b = -e_t$) gives

$$\langle \nabla f(x_t), s_t \rangle = -\eta \langle \nabla f(x_t), m_t \rangle_{D_t} = -\tfrac{\eta}{2} \|\nabla f(x_t)\|^2_{D_t} - \tfrac{\eta}{2} \|m_t\|^2_{D_t} + \tfrac{\eta}{2} \|e_t\|^2_{D_t}.$$

From $m_t = -\frac{1}{\eta} D_t^{-1} s_t$ we get $\frac{\eta}{2} \|m_t\|^2_{D_t} = \frac{1}{2\eta} \|s_t\|^2_{D_t^{-1}}$. Substituting,

$$f(x_{t+1}) \leq f(x_t) - \tfrac{\eta}{2} \|\nabla f(x_t)\|^2_{D_t} - \tfrac{1}{2\eta} \|s_t\|^2_{D_t^{-1}} + \tfrac{\eta}{2} \|e_t\|^2_{D_t} + \tfrac{L}{2} \|s_t\|^2.$$

Invoking A5 three times pathwise—$\|\nabla f(x_t)\|^2_{D_t} \geq \mu \|\nabla f(x_t)\|^2$, $\|e_t\|^2_{D_t} \leq M \|e_t\|^2$, and $\|s_t\|^2 \leq M\|s_t\|^2_{D_t^{-1}}$ (from $I \preceq M D_t^{-1}$)—and collecting the two $\|s_t\|^2_{D_t^{-1}}$ terms yields (37). ($D_t$ is random and correlated with $m_t, e_t$, but (37) is a pathwise inequality and the A5 bounds hold almost surely.) $\square$

The first-moment recursion (30) is identical to the main text, and $D_t$ enters the contraction analysis only through the displacement $\|x_t - x_{t-1}\|^2$. Hence the momentum-contraction bound is *exactly* Lemma B.2: for every $t \geq 1$,

$$\mathbb{E} \|e_t\|^2 \leq \left(1 - \beta_1\left(1 - \tfrac{\delta}{2}\right)\right) \mathbb{E} \|e_{t-1}\|^2 + \frac{2L^2}{\delta\beta_1} \mathbb{E} \|x_t - x_{t-1}\|^2 + (\beta_1 + 6\beta_1^2)(1-\delta) \mathbb{E} \|\nabla f(x_t)\|^2 + 3(2-\delta)\beta_1^2\sigma^2. \tag{38}$$

*Proof of Theorem N.2.* Write $\Delta_0 := f(x_0) - \inf_x f$, $e_0 := m_0 - \nabla f(x_0)$, $s_t := x_{t+1} - x_t$.

**Step 1 (sum the descent lemma).** Summing (37) over $t = 0, \ldots, T$, taking expectations and using $\mathbb{E} f(x_{T+1}) \geq \inf_x f$,

$$\frac{\eta\mu}{2} \sum_{t=0}^{T} \mathbb{E} \|\nabla f(x_t)\|^2 + \left(\frac{1}{2\eta} - \frac{LM}{2}\right) \sum_{t=0}^{T} \mathbb{E}\|s_t\|^2_{D_t^{-1}} \leq \Delta_0 + \frac{\eta M}{2} \sum_{t=0}^{T} \mathbb{E} \|e_t\|^2. \tag{39}$$

**Step 2 (telescope the contraction lemma).** Writing (38) as $\mathbb{E} \|e_t\|^2 \leq c\, \mathbb{E} \|e_{t-1}\|^2 + a_t$ with $c := 1 - \beta_1(1 - \tfrac{\delta}{2})$ and summing over $t = 1, \ldots, T$, then using $1 - c = \beta_1(1 - \tfrac{\delta}{2})$ and the bounds $\frac{1}{1-\delta/2} \leq 2$, $\frac{1-\delta}{1-\delta/2} \leq 1 - \tfrac{\delta}{2}$, $\frac{2-\delta}{1-\delta/2} = 2$,

$$\sum_{t=0}^{T} \mathbb{E} \|e_t\|^2 \leq \frac{2\mathbb{E} \|e_0\|^2}{\beta_1} + \frac{4L^2}{\delta\beta_1^2} \sum_{t=1}^{T} \mathbb{E} \|x_t - x_{t-1}\|^2 + \left(1 - \tfrac{\delta}{2}\right)(1 + 6\beta_1) \sum_{t=1}^{T} \mathbb{E} \|\nabla f(x_t)\|^2 + 6T\beta_1\sigma^2. \tag{40}$$

Converting the displacement to the variable metric via A5 ($\|x_t - x_{t-1}\|^2 = \|s_{t-1}\|^2 \leq M\|s_{t-1}\|^2_{D_{t-1}^{-1}}$) and extending the sums to $t = 0, \ldots, T$,

$$\sum_{t=0}^{T} \mathbb{E}\,\|e_t\|^2 \leq \frac{2\mathbb{E}\,\|e_0\|^2}{\beta_1} + \frac{4L^2 M}{\delta\beta_1^2} \sum_{t=0}^{T} \mathbb{E}\|s_t\|^2_{D_t^{-1}} + \left(1 - \frac{\delta}{2}\right)(1 + 6\beta_1) \sum_{t=0}^{T} \mathbb{E}\,\|\nabla f(x_t)\|^2 + 6T\beta_1\sigma^2. \quad (41)$$

**Step 3 (substitute and isolate).** Inserting (41) into (39) and regrouping,

$$\frac{\eta}{2}\Big[\mu - M\big(1 - \tfrac{\delta}{2}\big)(1 + 6\beta_1)\Big] \sum_{t=0}^{T} \mathbb{E}\,\|\nabla f(x_t)\|^2 + \Big[\frac{1}{2\eta} - \frac{LM}{2} - \frac{2\eta L^2 M^2}{\delta\beta_1^2}\Big] \sum_{t=0}^{T} \mathbb{E}\|s_t\|^2_{D_t^{-1}}$$

$$\leq \Delta_0 + \frac{\eta M}{\beta_1} \mathbb{E}\,\|e_0\|^2 + 3\eta M T\beta_1\sigma^2. \quad (42)$$

**Step 4 (apply the two conditions).** By (34), $M(1 - \tfrac{\delta}{2})(1 + 6\beta_1) \leq \mu(1 - \tfrac{\delta}{4})$, so the gradient coefficient obeys $\mu - M\big(1 - \tfrac{\delta}{2}\big)(1 + 6\beta_1) \geq \frac{\mu\delta}{4}$, whence $\frac{\eta}{2}[\cdot] \geq \frac{\eta\mu\delta}{8}$. The step-size condition makes both halves of the step coefficient nonnegative ($\eta \leq \frac{1}{2LM} \Rightarrow \frac{1}{4\eta} \geq \frac{LM}{2}$, and $\eta \leq \frac{\beta_1}{LM}\sqrt{\delta/8} \Rightarrow \frac{1}{4\eta} \geq \frac{2\eta L^2 M^2}{\delta\beta_1^2}$), so the nonnegative step-displacement sum may be dropped. Hence

$$\frac{\eta\mu\delta}{8} \sum_{t=0}^{T} \mathbb{E}\,\|\nabla f(x_t)\|^2 \leq \Delta_0 + \frac{\eta M}{\beta_1} \mathbb{E}\,\|e_0\|^2 + 3\eta M T\beta_1\sigma^2,$$

and dividing by $\frac{\eta\mu\delta(T+1)}{8}$ with $\frac{T}{T+1} \leq 1$ gives (35). □

*Proof of Corollary N.3.* With $\beta_1 = \theta/\sqrt{T+1} \to 0$, the left side of (34) tends to $\kappa_D(1 - \tfrac{\delta}{2}) < 1 - \tfrac{\delta}{4}$ by the strict assumption on $\kappa_D$, so (34) holds for large $T$; the choice $\eta = \frac{\beta_1}{LM}\sqrt{\delta/8}$ satisfies the step-size condition. Substituting into (35) and using $\beta_1(T+1) = \theta\sqrt{T+1}$, every term carries a common factor $1/\sqrt{T+1}$, giving (36). □

**Scope and relation to the Adam literature.** The whole momentum-error algebra (Lemma B.2) is reused unmodified; the only new object is the variable-metric descent lemma and the only new hypothesis is the spectral bound A5, which is why the statement collapses exactly onto Theorem 4.5 at $\kappa_D = 1$. Condition (34) forces $\kappa_D < \frac{4-\delta}{4-2\delta} \in (1, \tfrac{3}{2}]$, i.e. a well-conditioned preconditioner (a non-negligible numerical floor $\varepsilon$, or the late-training phase in which $v_t$ has stabilized). A5 is the minimal addition that preserves the manuscript's Euclidean descent-and-telescope machinery; it is not the weakest known sufficient condition for Adam. Sharper optimizer-specific analyses remove it under different techniques—$\beta_1 < \sqrt{\beta_2}$ with large $\beta_2$ (Zhang et al., 2022), bounded-gradient $O(\log T/\sqrt{T})$ bounds (Défossez et al., 2022), controlled effective-step oscillation (Chen et al., 2019). Our pretraining runs use $\beta_1^{\text{Adam}}=0.9$, $\beta_2=0.95$, consistent with the large-$\beta_2$, heavy-momentum regime in which both our result and Zhang et al. (2022) apply.

## O   System Evaluation: Latency, Jitter, Heterogeneity, and Pipeline Depth

The main text establishes throughput gains under throttled bandwidth. We now characterize the remaining system axes relevant to a decentralized deployment: network latency, jitter, heterogeneous participants, pipeline depth, and multi-node scale beyond 8 GPUs.

**Setup.** We use two nodes of 8 H20 GPUs linked over a TCP socket with InfiniBand disabled, emulating a commodity decentralized link. The workload is `GPT-2XL` under pipeline parallelism at sequence length 1024. The uncompressed baseline transmits FP16 activations in both directions, whereas TAH-QUANT transmits the forward tile-wise adaptive Hadamard activations with a 3.8-bit payload (about 4.4 bit/element including worst-case metadata; Appendix E) together with a 6-bit fixed-point backward. We interleave the pipeline

stages so that consecutive stages sit on different nodes, making every pipeline boundary cross the network and reproducing the paper's geo-distributed topology on two physical machines. The inter-node link is shaped with Linux `tc`, filtered to the peer node so that storage and control traffic stay untouched: `netem` delay for latency, a rate-limited delay-plus-jitter band for jitter, and `nvidia-smi` clock capping for heterogeneity. Throughput is reported as tokens per second, averaged over steady-state iterations.

Table 18: System evaluation: TAH-QUANT throughput speedup over the uncompressed FP16 baseline. Unless noted, PP=8 with every boundary crossing the network.

(a) Latency sweep (PP8).

| added latency | none | TAH-QUANT | speedup |
|---|---|---|---|
| 0 (LAN) | 7940 | 8201 | 1.03× |
| 50 ms | 1788 | 3454 | 1.93× |
| 100 ms | 576 | 2006 | 3.48× |

(b) Pipeline depth / scale (@ 50 ms).

| depth | GPUs | none | TAH-QUANT | speedup |
|---|---|---|---|---|
| PP4 | 4 | 2328 | 3649 | 1.57× |
| PP8 | 8 | 1771 | 3454 | 1.95× |
| PP16 | 16 | 1151 | 2191 | 1.90× |

(c) Heterogeneity (@ 50 ms, PP8; one node ≈2.2× slower).

| setting | none | TAH-QUANT | speedup |
|---|---|---|---|
| homogeneous | 1788 | 3454 | 1.93× |
| heterogeneous | 1713 | 2740 | 1.60× |

(d) Jitter (PP8, 300 Mbps, 50±20 ms) and overhead.

| setting | none | TAH-QUANT | speedup |
|---|---|---|---|
| 50±20 ms jitter | 403 | 2076 | 5.15× |
| compute overhead | 3.263 | 3.349 | +2.66% |

**Findings.** Table 18 summarizes the results. TAH-QUANT is already faster than the uncompressed baseline on a LAN at 1.03×, and the speedup grows with latency to 1.93× at 50 ms and 3.48× at 100 ms, where the baseline collapses to 576 tok/s while TAH-QUANT holds 2006. The depth sweep covers both alternative pipeline depths and multi-node scale beyond 8 GPUs: PP16 over 16 GPUs runs cleanly, and the speedup holds as depth grows, reaching 1.57×, 1.95×, and 1.90× at PP4, PP8, and PP16 while the baseline falls as boundaries multiply. A 2.2× compute straggler narrows the gain but does not erase it, leaving 1.60×, and the roughly 4× smaller payload makes TAH-QUANT far more robust to jitter at 5.15×. We read the jitter result as robustness rather than a precise ratio, since TCP reordering under jitter inflates the magnitude. The operator adds only 2.66% compute on a single node with no network limit, consistent with the ∼1% of Section 3.4. We do not evaluate participant failures such as node drop and recovery, which are orthogonal to the compression contribution.

## P  Held-out Perplexity and Equal-bit Ablation for `GPT-2XL` Fine-tuning

**Held-out perplexity.** Beyond training loss, we report held-out perplexity (Table 19) for the `GPT-2XL` fine-tuning arms on two corpora (same configuration as the main text). TAH-QUANT (which transmits ≈ 4.4 bit/element; Appendix E) is indistinguishable from the uncompressed baseline and from the higher-precision uniform `fixed-INT4` reference, while the 3-bit arm is measurably worse, confirming that the low-bit forward compression preserves out-of-sample quality, not just training loss. We also include the `AQ-SGD` baseline where it is feasible (the multi-epoch `WikiText` setting); it converges to the same held-out perplexity as TAH-QUANT and `FP16`, but at higher per-step communication cost, so TAH-QUANT reaches a given perplexity faster in wall-clock time (Section 5.2).

**Equal-bit ablation.** Table 20 isolates each component of TAH-QUANT under a controlled `GPT-2XL`/`WikiText` setup (∼700 steps, PP=4, 3 seeds). Every arm uses the TAH-QUANT operator and disables exactly one ingredient while holding the activation precision at TAH-QUANT's 3.8-bit payload (≈ 4.4 bit/element transmitted; Appendix E); the `fixed-INT4` and `fixed-INT3` rows are uniform 4-/3-bit references. The arms thus isolate the effect of fixed (non-adaptive) precision, removing the pivot swap, removing the Hadamard transform, applying the transform to all tiles, and token-level (rather than tile-wise) allocation.

Each component contributes positively, and the ordering is stable across all three seeds. The Hadamard transform is the dominant component (the largest degradation when removed), selective application ($\tau$=2)

Table 19: Held-out perplexity of the `GPT-2XL` fine-tuning arms (lower is better).

| Arm | WikiText PPL | ArXiv21 PPL |
|---|---|---|
| uncompressed `FP16` | 12.97 | 15.45 |
| **TAH-Quant** | 12.97 | 15.48 |
| `AQ-SGD` | 12.97 | — |
| `fixed-INT4` (uniform 4-bit) | 12.96 | 15.48 |
| `fixed-INT3` (uniform 3-bit) | 13.04 | 15.56 |

Table 20: Equal-bit component ablation on `GPT-2XL`/WikiText (mean final training loss over 3 seeds; lower is better).

| Arm | final loss |
|---|---|
| `fixed-INT4` (uniform 4-bit) | 2.467 |
| **TAH-Quant (adaptive 4/3)** | **2.476** |
| TAH-QUANT, no adaptive allocation | 2.483 |
| TAH-QUANT, no pivot swap | 2.485 |
| token-level allocation ($A=C$) | 2.486 |
| Hadamard on all tiles ($\tau=0$) | 2.499 |
| `fixed-INT3` (uniform 3-bit) | 2.513 |
| TAH-QUANT, no Hadamard ($\tau=\infty$) | 2.528 |

beats applying it to all tiles, and TAH-QUANT matches the higher-precision uniform `fixed-INT4` within noise. A simple **per-channel** quantizer is far worse at every bit-width (held-out WikiText PPL 13.31/15.26/39.10 at 4/3/2-bit vs. TAH-QUANT 12.97), confirming that fine-grained tiling is the prerequisite for low-bit forward activation quantization.

# Q   AQ-SGD Cache Overhead and Applicability

`AQ-SGD` reduces activation-communication error by transmitting the quantized *difference* between the current activation and the same example's activation cached from the previous epoch. This requires a per-example activation cache that persists across epochs: for every training example, its activation at each pipeline cut is stored so that the error-compensation delta can be formed when the example is revisited. The cache size is

$$\text{cache} \;=\; N_{\text{ex}} \times (P-1) \times S \times C \times (\text{bytes/elem}), \tag{43}$$

where $N_{\text{ex}}$ is the number of training examples, $P$ the pipeline depth ($P-1$ inter-stage cuts), $S$ the sequence length, and $C$ the hidden width. For our `GPT-2XL` configuration ($S=1024$, $C=1600$), one cached activation per example per cut is $S \cdot C=1.64$M values (3.28 MB at FP16), i.e. $\approx 23$ MB per example at PP=8 (7 cuts). Table 21 reports the resulting full-dataset cache across the evaluated corpora (normalized to 1024-token sequences using this representative activation size).

Table 21: `AQ-SGD` full-dataset activation-cache size and applicability across the evaluated tasks. The cache grows linearly with dataset size, and `AQ-SGD` additionally requires each example to recur across epochs to form its error-compensation delta.

| Dataset (task) | $\approx \# 1024$-tok seqs | cache @ FP16 | `AQ-SGD` applicable? |
|---|---|---|---|
| `WikiText-2` (i) | 2.4K | $\approx 55$ GB | yes (multi-epoch) |
| `Open-Platypus` (iv) | 25K | $\approx 0.57$ TB | no (single epoch) |
| `Magicoder-110K` (iii) | 110K | $\approx 2.5$ TB | no (storage-prohibitive) |
| `SlimPajama-6B` (v) | 5.9M (single pass) | $\approx 134$ TB | no (single pass) |

Two independent conditions govern applicability. **(i) Storage.** The cache grows linearly with dataset size, reaching multiple terabytes for the instruction-tuning and pretraining corpora—prohibitive to store. **(ii) Reuse.** The error-compensation delta is defined only when an example is revisited, so under one-pass/single-epoch training (`Open-Platypus`, and the from-scratch `SlimPajama-6B` pretraining) the cached activation is never reused and the mechanism provides no benefit in its intended form. Only a multi-epoch task with a modest corpus—`WikiText-2` ($\approx 55$ GB)—satisfies both, which is exactly where we run the head-to-head comparison: `AQ-SGD`, TAH-Quant, and `FP16` all reach the same held-out perplexity (12.97; Appendix P), while `AQ-SGD` additionally pays a higher per-step communication and activation-cache I/O cost. By contrast, TAH-Quant is a stateless, on-the-fly quantizer with *zero* activation cache and therefore applies uniformly across the fine-tuning, instruction-tuning, and from-scratch pretraining settings.

## R    Comparison with Subspace Networks

A complementary line of communication-efficient model parallelism reduces inter-stage traffic by constraining the model parameterization rather than compressing activations. We compare against **Subspace Networks** Ramasinghe et al. (2025), which projects the projection-layer weights onto a $k$-dimensional subspace. We match the forward communication volume per pipeline boundary: Subspace at $k=400$ (from $d=1600$) and TAH-Quant at 4-bit both achieve $\sim 4\times$ *nominal* forward compression (TAH-Quant transmits a 3.8-bit payload, $\approx 4.43$ bit/element including worst-case metadata; Appendix E). Both arms use FP16 parameters and activations with an 8-bit fixed-point backward.

Table 22: Fine-tuning `GPT-2XL` on `WikiText` (PP=8): training loss at several steps.

| Method | compr. | @100 | @200 | @400 | @600 |
|---|---|---|---|---|---|
| Uncompressed | $1\times$ | 2.683 | 2.595 | 2.505 | 2.467 |
| Subspace ($k=1600$) | $\sim 1\times$ | 2.683 | 2.597 | 2.509 | 2.470 |
| Subspace ($k=400$) | $\sim 4\times$ | 7.728 | 6.987 | 5.987 | 5.688 |
| **TAH-Quant (4-bit)** | $\sim 4\times$ | 2.694 | 2.607 | 2.519 | 2.482 |

Table 23: From-scratch pretraining on `ArXiv21` (PP=4): training loss at several steps.

| Method | compr. | @100 | @200 | @500 | @1000 |
|---|---|---|---|---|---|
| Uncompressed | $1\times$ | 6.573 | 5.652 | 4.672 | 3.826 |
| Subspace ($k=400$) | $\sim 4\times$ | 6.804 | 5.659 | 4.662 | 3.861 |
| **TAH-Quant (4-bit)** | $\sim 4\times$ | 6.570 | 5.666 | 4.688 | 3.858 |
| Subspace + TAH-Quant (6-bit) | $\sim 10\times$ | 6.803 | 5.663 | 4.661 | 3.861 |

At matched $\sim 4\times$ compression (Table 22), TAH-Quant tracks the uncompressed run (gap $< 0.02$), whereas Subspace at $k=400$ degrades substantially (5.69 vs. 2.47). Constraining a *pretrained, full-rank* model to a 400-dimensional subspace discards most of its learned directions; Subspace at $k=1600$ (no rank reduction) matches uncompressed, confirming the degradation is inherent to low-rank projection of pretrained weights.

When training from scratch (Table 23) there are no pretrained directions to lose, and both methods converge comparably at $\sim 4\times$ (within 0.035 of uncompressed). Because the two compress from orthogonal angles—low-rank weight projection versus activation quantization—they compose: combining Subspace with TAH-Quant reaches $\sim 10\times$ compression with virtually identical convergence. Subspace Networks is therefore complementary to TAH-Quant rather than competing.

