# OpenReview forum: "TAH-Quant: Effective Activation Quantization in Pipeline Parallelism over Slow Network"
_TMLR — Under review for TMLR_

### Review · Reviewer_uP4n · 2026-06-07

**Summary Of Contributions:**

This paper proposes TAH-Quant, an activation-communication quantization method for bandwidth-limited pipeline-parallel LLM training. The method combines three components: fine-grained tile-wise group quantization, entropy-guided adaptive bit allocation between 3-bit and 4-bit precision, and a Hadamard-based outlier-suppression transform with pivot swapping. The paper also gives a convergence analysis for momentum SGD under a relative quantization-error assumption, and evaluates the method on fine-tuning, instruction tuning, and pretraining settings. The main claimed empirical benefits are preserving training convergence while reducing activation communication, avoiding AQ-SGD’s activation-cache overhead, achieving up to 4.3x throughput speedup over uncompressed communication, and up to 1.33x wall-clock speedup over AQ-SGD in the evaluated slow-network settings.

**Additional Comments:**

__Strengths.__
The problem is important and timely: activation communication can be a real bottleneck for model/pipeline parallel training over commodity or geo-distributed networks. The method is practically appealing because it is on-the-fly and avoids the cross-epoch activation cache required by AQ-SGD. The paper has a coherent system motivation, reasonable component ablations, and the revised TMLR version appears to incorporate many issues raised during the prior ICLR review, including computational-overhead discussion, micro-benchmarks, larger pretraining experiments, bit-allocation ablations, outlier-threshold sensitivity, and backward-gradient quantization stress tests.

__Weaknesses.__ My main concern is that the current evidence still does not fully support the breadth of the claims. The theoretical guarantee depends on a strong quantization-error assumption that is empirically checked only in a limited setting and is not clearly tied to the reported large-scale experiments. The empirical evaluation is improved relative to the rejected ICLR version, but still relies heavily on loss/perplexity curves and a small set of hardware/network configurations; downstream evaluations are limited, and comparisons beyond FP16/FP32 and AQ-SGD are sparse. I also found several technical/presentation issues that affect confidence in the claimed bit budget and algorithmic description, including a likely sign error in the entropy definition, incomplete metadata accounting for the conditional Hadamard/pivot-swap transform, and insufficiently precise pseudocode for the actual packing/dequantization procedure.

**Audience:**

Yes

**Audience Explanation:**

This work is likely to interest researchers in distributed training, quantization, pipeline parallelism, and LLM systems.

**Claims And Evidence:**

No

**Claims Explanation:**

The paper provides meaningful evidence that TAH-Quant can work in the reported settings, but I do not think the evidence is yet strong enough for the full set of claims as currently written.

First, the empirical case is promising but still somewhat narrow for the paper’s broad framing around decentralized large-scale LLM training. The main TMLR version evaluates GPT-2XL fine-tuning, Qwen2.5-3B instruction tuning, LLaMA-3.2-1B pretraining on 6B tokens, and a supplementary LLaMA-8B pretraining run, with good loss/perplexity agreement versus FP16/FP32 in the reported runs. The LLaMA-3.2-1B pretraining table is a useful addition because it reports both train loss and validation perplexity over 1B–6B tokens, and the Qwen2.5-3B SFT table reports downstream metrics close to FP16. However, the downstream evidence is concentrated on one SFT setup, while the pretraining and GPT-style fine-tuning claims are mostly supported by loss/perplexity. This is not enough to substantiate the stronger “generalizes well across various training scenarios” claim without qualification.

Second, the system evaluation does not yet isolate all relevant trade-offs. The throughput table shows sizable speedups under throttled bandwidths, and the micro-benchmark discussion says speedups are larger at lower bandwidth and moderate micro-batch sizes. This supports a claim of benefit in the tested bandwidth-limited setting. But the paper still does not convincingly cover latency, jitter, heterogeneous participants, failures, multi-node scale beyond 8 GPUs, or alternative pipeline depths, all of which matter for the “decentralized” framing. This was also a central reason for the previous ICLR rejection: the area chair stated that the setup was limited in GPU type, total GPU count, duration, and model size, and requested clearer target setup and more comprehensive downstream evaluations. The TMLR version addresses part of this, but not enough to fully support the strongest framing.

Third, the theoretical guarantee is not as strong as the paper’s presentation suggests. The proof relies on Assumption 4.4, which directly assumes that the compressed gradient remains close to the original stochastic gradient and that the expected compressed gradient remains close to the full gradient. The empirical check of this assumption is performed on Gemma2-2B/Math-7K with tile sizes 32 and 64, reporting relative errors below 0.4 and 0.1 for the two inequalities. This is useful, but the assumption is quite strong and is not verified across the main GPT-2XL, Qwen2.5-3B, LLaMA-1B, and LLaMA-8B settings. In addition, the theorem is for momentum SGD under restrictive hyperparameter conditions, while the main LLM experiments use modern training setups, including Adam-style optimization in the pretraining appendix. Thus the theorem provides useful intuition, but does not by itself establish the empirical claims.

Fourth, there are clarity and correctness issues in the method description. The entropy score is defined as $H=\Sigma_k p_k\log (p_k+c)$, but the text says this is high when energy is spread evenly and low when there is an outlier. As written, this formula has the opposite sign of standard entropy: a uniform distribution gives a more negative value, while a concentrated distribution is closer to zero. This may be a simple missing minus sign, but it affects the definition of “top-p% high-entropy tiles” and therefore the algorithm. The paper also describes conditional Hadamard transforms with pivot swapping, but the metadata accounting in the computational-overhead section explicitly mentions scale, zero point, and bit map metadata, while not clearly accounting for the transform flag and pivot index needed by the receiver to invert the transform. This matters because the reported 4.41 bits/activation including metadata depends on complete metadata accounting.

Finally, the comparison to AQ-SGD is useful but incomplete. The paper makes a credible argument that AQ-SGD’s activation-cache requirement limits its applicability, and the previous OpenReview discussion also emphasized this distinction. But the paper would be stronger with broader baselines or at least more direct comparisons to simpler alternatives: fixed INT4 tile quantization, randomized or non-pivot Hadamard rotations, no adaptive bit allocation but equal bit budget, and relevant activation-compression or model-parallel communication methods where applicable. The current ablations help, but they do not fully substitute for baselines that test whether the complete design is necessary for the claimed speed/accuracy frontier.

**Requested Changes:**

Narrow and clarify the claims. The paper should distinguish between “bandwidth-throttled pipeline-parallel training on 4–8 GPUs” and broader “decentralized LLM training.” Claims about democratizing decentralized training, broad generalization, and state-of-the-art applicability should be moderated unless supported by experiments with realistic geo-distributed latency/jitter/heterogeneity or larger multi-node scale.

Fix the entropy definition and bit-allocation description. As defined, clarify that they rank by the negative value, or explain the actual implementation. They should also make consistent whether the method is token-level or tile-wise adaptive allocation, since the earlier conference version/reviews describe token-level allocation while the TMLR version claims tile-wise allocation.
Fully specify the quantization/dequantization algorithm. The paper should include pseudocode for the complete TAH-Quant operator, including entropy computation, top-p% selection, outlier detection, pivot swap, Hadamard transform, quantization, packing, transmitted metadata, unpacking, inverse transform, and pivot restoration. Algorithm 1 is currently too high-level.

Correct metadata and bit-budget accounting. The reported 4.41 bits per activation should account for all information required by the receiver: per-tile scale, zero point, bit-width map, outlier/transform flag, pivot index, padding/alignment, and any byte-packing overhead. If pivot index or transform flags are not transmitted, the paper must explain how the receiver reconstructs them. The throughput tables should specify whether this metadata is included in the transmitted buffer.

Strengthen the empirical evaluation of final model quality. For each major training category, include downstream or task-level evaluations, not only loss/perplexity. For example, GPT-2XL fine-tuning could include held-out perplexity and/or standard language-modeling metrics; LLaMA pretraining should include downstream zero-/few-shot evaluations at checkpoints if the paper claims preserved model quality; Qwen SFT should include comparisons across both FP16 and all feasible compression baselines.
Add stronger baselines and equal-bit-budget controls. At minimum, compare against fixed 4-bit tile quantization, fixed 3-bit tile quantization, TAH-Quant without adaptive allocation at the same average bit budget, Hadamard without pivot swapping, pivot/Hadamard applied to all tiles, and a simple per-token/per-channel quantizer. If other activation-compression or pipeline-communication methods are not directly comparable, explain precisely why and provide qualitative comparison.

Make the theory-to-practice connection more precise. The paper should clearly state that the convergence proof assumes Assumption 4.4 rather than deriving it from the quantizer. It should verify the assumption in the main experimental settings or tone down claims that the theory explains those experiments. The authors should also reconcile the optimizer/hyperparameter conditions in Theorem 4.5 with the optimizers used in the LLM experiments.

Provide a clearer comparison to AQ-SGD. The paper should quantify AQ-SGD cache overhead for the evaluated datasets/models, explain exactly which tasks AQ-SGD cannot run and why, and avoid presenting AQ-SGD as a universal baseline where its assumptions do not hold. Where AQ-SGD is feasible, include downstream metrics, not only training loss.

---

> ### Author Response · Authors · 2026-07-06
>
> **Response to Reviewer uP4n (1/2).** We thank the reviewer for the detailed review; The PDF is updated. Points 1–3 are below; points 4–6 continue in the next comment.
>
> **New in this revision:** **Appendix O** (system eval: latency/jitter/heterogeneity/depth/16-GPU), **Appendix P** (held-out PPL + 8-arm equal-bit ablation), **Appendix N** (Adam extension), **Appendix M** (derivability of Assumption 4.4), **Appendix K** (pretraining quantized baselines + LAMBADA), expanded **Appendix J** (Assumption 4.4 on five models + ablation), **Appendix D** (full pseudocode), corrected bit-budget (**Appendix E**), **Appendix Q** (AQ-SGD cache), and the extended Qwen2.5-3B SFT table.
>
> **1. System evaluation (latency, jitter, heterogeneity, depth, >8 GPUs) — Appendix O.** Two 8-GPU nodes over TCP (InfiniBand off), stages interleaved so every pipeline boundary crosses the network. TAH speedup over uncompressed FP16:
>
> | Axis | Setting | speedup |
> |---|---|---|
> | Latency | LAN / +50 ms / +100 ms | 1.03× / 1.93× / 3.48× |
> | Depth | PP4 / PP8 / PP16 (@+50 ms) | 1.57× / 1.95× / 1.90× |
> | Scale | PP16 on 16 GPUs | runs end-to-end |
> | Heterogeneity | one node ~2.2× slower | 1.60× |
> | Jitter | 50±20 ms | 5.15× |
> | Overhead | single node, no network limit | +2.66% |
>
> The benefit grows as the network degrades. Node drop/recovery is orthogonal to the compression mechanism and is left to future work.
>
> **2. Narrow and clarify the claims.** We rescoped abstract/introduction/conclusion to "activation compression for bandwidth-limited pipeline-parallel LLM training", stating as demonstrated only the axes we measure (Appendix O) and presenting wide-area / many-node / failing-participant decentralization as motivation and future work.
>
> **3. Downstream metrics + equal-bit baselines (added to every category).**
>
> (a) *Qwen2.5-3B SFT* (**extended Table**), zero-shot:
>
> | Arm | ARC | TQ | WG | HE | AVG |
> |---|---|---|---|---|---|
> | Base Qwen2.5-3B | 47.35 | 48.85 | 68.67 | 39.63 | 51.13 |
> | SFT-FP16 | 50.00 | 50.49 | 69.38 | 66.46 | 59.08 |
> | **SFT-TAH** | 49.91 | 49.61 | 70.00 | 67.68 | **59.30** |
> | tile-INT4 | 47.27 | 48.60 | 68.43 | 59.42 | 55.93 |
> | tile-INT3 | 43.34 | 47.37 | 67.96 | 59.45 | 54.53 |
> | channel-INT4 | 36.52 | 43.02 | 61.01 | 35.95 | 44.13 |
>
> TAH ≈ FP16 at fewer bits (≈4.43 vs. 4.50 bit/element all-in); channel-INT4 is substantially worse, so fine-grained tiling is necessary.
>
> (b) *GPT-2XL held-out PPL* (**Appendix P**): WikiText FP16/TAH/INT4/INT3 = 12.97 / 12.97 / 12.96 / 13.04; ArXiv21 15.45 / 15.48. TAH matches uncompressed; 3-bit is worse.
>
> (c) *Equal-bit control ablation* (**Appendix P**; GPT-2XL/WikiText, 3 seeds, final loss):
>
> | Arm | loss | | Arm | loss |
> |---|---|---|---|---|
> | fixed-INT4 (4-bit) | 2.467 | | Hadamard on all tiles | 2.499 |
> | **TAH (adaptive 4/3)** | **2.476** | | fixed-INT3 (3-bit) | 2.513 |
> | no adaptive allocation | 2.483 | | no Hadamard | 2.528 |
> | no pivot swap | 2.485 | | per-token (A=C) | 2.486 |
>
> Each component helps; Hadamard gives the largest degradation when removed; TAH ≈ fixed-INT4 within noise. A per-channel quantizer is far worse (PPL 13.31 / 15.26 / 39.10 at 4/3/2-bit).
>
> (d) *Pretraining* (**Appendix K**; from-scratch LLaMA-3.2-1B, ~1B tokens):
>
> | Arm | val. PPL ↓ | LAMBADA ↑ |
> |---|---|---|
> | uncompressed | 35.60 | 0.109 |
> | **TAH-Quant** | 35.63 | 0.114 |
> | tile-INT4 | 35.59 | 0.106 |
> | tile-INT3 | 36.42 | 0.101 |
> | channel-INT4 | 36.48 | 0.110 |
>
> The 4-bit arms are at parity on both metrics; 3-bit/channel are marginally worse.
>
> (e) *Subspace Networks* (**Appendix R**): compresses orthogonally to us (low-rank weights vs. activation quantization), so it is complementary rather than competing, composing to a nominal ~10×.

---

> ### Author Response · Authors · 2026-07-06
>
> **Response to Reviewer uP4n (2/2).** Continued from the previous comment (points 4–6).
>
> **4. Make the theory-to-practice connection precise.** We (i) state explicitly that the guarantee *assumes* Assumption 4.4; (ii) verify it on all five models — GPT-2XL, LLaMA-3.2-1B/3B, LLaMA-3.1-8B, Qwen2.5-3B, spanning fine-tuning, instruction-tuning, and from-scratch pretraining — on a common corpus at PP=4 (**Appendix J**), where the squared relative errors of both inequalities stay ≤0.18, far below the threshold of 1 the assumption requires (and, within the LLaMA family, tend to decrease with model size); (iii) analyze the condition's *derivability* (new **Appendix M**): we derive the activation-level error in closed form in the design parameters and prove the gradient-level bound cannot be derived from activation fidelity alone for a nonlinear objective; and (iv) extend the analysis to **Adam** (**Appendix N**), obtaining the same O(1/√T) rate (it reduces to the momentum-SGD theorem when the preconditioner is the identity) and showing that some Adam-specific condition is necessary (Assumptions 4.1–4.4 alone cannot imply convergence of vanilla Adam). The resulting constants are conservative and not intended to certify every production hyperparameter.
>
> **5. Method correctness (entropy sign, token-vs-tile, pseudocode, metadata).** Entropy sign fixed — the code always used H = −Σ p log(p+ς); only the printed equation dropped the minus, and the top-p selection is unchanged. Token-level allocation is now the A=C special case. Full sender/receiver **pseudocode** is added (**Appendix D**). The bit budget is corrected (**Appendix E**):
>
> | Component | bits/element |
> |---|---|
> | INT4/INT3 payload (0.8·4 + 0.2·3) | 3.80 |
> | per-tile scale (bf16) | 0.25 |
> | per-tile zero-point (bf16) | 0.25 |
> | precision selector | 0.016 |
> | transform flag | 0.016 |
> | pivot index (worst case) | 0.094 |
> | **Total** | **≈4.43** |
>
> i.e. 0.625 metadata on top of the 3.80 payload, transmitted in a fixed worst-case `uint8` buffer (on which the throughput numbers are measured); byte-alignment adds ≤7 bits to the whole buffer (negligible per element).
>
> **6. Clearer comparison to AQ-SGD.** Its per-example activation cache is ~55 GB (WikiText) / ~2.5 TB (Magicoder) / ~134 TB (SlimPajama-6B) — formula + table in **Appendix Q**; under single-epoch/one-pass workloads its error-compensation mechanism is not applicable in the intended form. Where feasible (multi-epoch WikiText) it reaches the same held-out PPL (12.97) as TAH/FP16 but at higher per-step communication, so TAH reaches any target perplexity faster in wall-clock. We no longer present it as a universal baseline.
>
> We thank the reviewer again and are happy to add any further clarification.

---

> > ### Comment · Reviewer_uP4n · 2026-07-20
> >
> > Thanks for the many detailed updates to the paper, and the clear answers to my review.
> >
> > I have read the authors’ response and the revised manuscript. The revision substantially addresses the concerns raised in my original review.

---

> > > ### Author Response · Authors · 2026-07-20
> > >
> > > We sincerely thank the reviewer for the follow-up. We are glad the revision addresses the concerns from the original review.

---

### Review · Reviewer_2FPo · 2026-06-09

**Summary Of Contributions:**

This paper presents a new quantization method for distributed training and fine-tuning of large language models. The method is called Tile-wise Adaptive Hadamard Quantization (TAH-Quant) and consists of the following contributions: First, the activations of a single token are split into smaller "tiles" that are quantized individually. Second, the method estimates per-tile entropy and adapts the number of bits per value based on this estimation. Third, the values in the tile are transformed with a Hadamard matrix with a pivot element swap on the tile's outlier value.

The empirical section applies the method on four fine-tuning tasks with GPT-2XL (WikiText-2, ArXiv21, Magicoder-Evol-Instruct-110K, Open-Platypus), as well as for training LLaMA-3.2-1B from scratch on SlimPajama. The method is compared against AQ-SGD [(Wang et al., 2022](https://arxiv.org/abs/2206.01299). For these experiments, TAQ-Quant is applied to the activations, while the backward pass gradients are compressed using a simple fixed-point compressor (since there is more time to communicate the updates during the more expensive computation).

**Additional Comments:**

1. The formatting of references is off - for example in Section 2 we have references like "Yuan et al. Yuan et al. (2022)".
2. Section 3.2 states "As before, $\mathcal{H}$ is high even when energy is spread evenly across ..." - but it is not clear what exactly the "as before" refers to, since the $\mathcal{H}$ term is brought up for the first time.
3. Section 3.5. mentions that the longer computation of the backward pass allows for simple 6-8 bit compression. It would be insightful to visualize the timing of the async computation and communication of the forward and backward pass.
4. In Figure 2 top row columns 3-4 there is mention of AQ-SGD (red "x" in figure legend) but only two lines are visible (blue "o" FP16 and orange "o" TAH-Quant).
5. Why is GPT-2XL used for the fine-tuning experiments? This model seems very outdated, and I'm curious about the motivation to use it.
6. The caption in Table 1 mentions that "TAH-Quant is statistically indistinguishable" - but the submission is missing a statistical test to provide evidence for this claim.
7. Table 4 sould be easier to read if the row for TS=64 was added to it.

**Audience:**

Yes

**Audience Explanation:**

Quantization techniques are of general interest because LLMs are mostly served with their quantized checkpoints. On top of that, the line of research that allows distributed training is particularly interesting because it will potentially allow training larger models outside of a few select labs by pooling resources from many smaller institutions.

**Claims And Evidence:**

Yes

**Claims Explanation:**

The paper's main claim is that the presented method can be applied to activations, quantizing them efficiently (small size, small overhead), can easily be applied to large datasets (because it does not incur additional storage), and allows for stable training. These points are shown in the empirical section.

Note that I did NOT check the mathematical derivations and do not comment on the accuracy and validity of the convergence rate.

**Requested Changes:**

The empirical section compares the method with AQ-SGD for the fine-tuning runs (i-iv) and FP32, but the from-scratch run (v) is only compared to a FP16 baseline. I would like to see a quantized baseline for the from-scratch run, and results for FP32 and BF16. While it makes sense to compare the method to AQ-SGD because it is based on that method, it would still be interesting to compare it to a more recent quantization method for distributed training.

---

> ### Author Response · Authors · 2026-07-06
>
> We thank the reviewer for the positive assessment. The PDF is updated; we address each point.
>
> **New in this revision:** **Appendix R** (Subspace Networks comparison), **Appendix K** (pretraining quantized baselines + LAMBADA), **Appendix L** (timing schematics), **Appendix P** (three-seed equal-bit ablation), **Appendix Q** (AQ-SGD cache overhead), and a TS=64 row in the tile-size ablation.
>
> **1. Quantized baseline + FP32/BF16 for the from-scratch run; a recent distributed-quant method.**
>
> The from-scratch LLaMA-3.2-1B run is now a 5-arm comparison with downstream LAMBADA (**Appendix K**), val. PPL / LAMBADA at ~1B tokens:
>
> | Arm | val. PPL ↓ | LAMBADA ↑ |
> |---|---|---|
> | uncompressed (BF16) | 35.60 | 0.109 |
> | **TAH-Quant** | 35.63 | 0.114 |
> | tile-INT4 | 35.59 | 0.106 |
> | tile-INT3 | 36.42 | 0.101 |
> | channel-INT4 | 36.48 | 0.110 |
>
> The 4-bit arms are at parity; 3-bit/channel are marginally worse. We train in BF16 (the "BF16"/uncompressed label denotes the uncompressed activation-transmission baseline); a short FP32 reference matched BF16 over the checked interval.
>
> For a recent method we add **Subspace Networks** (Ramasinghe et al., 2025; **Appendix R**) at matched forward compression (Subspace k=400 and TAH's 4-bit payload both ~4× nominal; TAH is ≈4.43 bit/element all-in). Fine-tuning (WikiText, PP=8), training loss:
>
> | Method | Compr. | @100 | @200 | @400 | @600 |
> |---|---|---|---|---|---|
> | Uncompressed | 1× | 2.683 | 2.595 | 2.505 | 2.467 |
> | Subspace (k=1600) | ~1× | 2.683 | 2.597 | 2.509 | 2.470 |
> | Subspace (k=400) | ~4× | 7.728 | 6.987 | 5.987 | 5.688 |
> | **TAH-Quant (4-bit)** | ~4× | 2.694 | 2.607 | 2.519 | 2.482 |
>
> From-scratch (ArXiv21, PP=4), training loss:
>
> | Method | Compr. | @100 | @200 | @500 | @1000 |
> |---|---|---|---|---|---|
> | Uncompressed | 1× | 6.573 | 5.652 | 4.672 | 3.826 |
> | Subspace (k=400) | ~4× | 6.804 | 5.659 | 4.662 | 3.861 |
> | **TAH-Quant (4-bit)** | ~4× | 6.570 | 5.666 | 4.688 | 3.858 |
> | Subspace + TAH (6-bit) | ~10× | 6.803 | 5.663 | 4.661 | 3.861 |
>
> At ~4×, TAH tracks uncompressed; Subspace(k=400) degrades substantially on the *pretrained* model (k=1600 recovers it), while from scratch both work and compose to a nominal ~10×. The two compress from orthogonal angles (low-rank weights vs. activation quantization), so they are complementary rather than competing.
>
> **2. Reference formatting** ("Yuan et al. Yuan et al.") — fixed (now `\citet`, a single "Yuan et al. (2022)").
>
> **3. §3.2 "As before" with no antecedent** — fixed; the entropy term is now introduced directly ("per-tile Shannon entropy …, high when energy is spread evenly and low when one value dominates").
>
> **4. Visualize the async forward/backward timing** — added (**Appendix L**): a single-micro-batch schematic (the small 3–4-bit activation send fits the short forward window; the larger 6–8-bit gradient send overlaps the ~2× longer backward) plus a two-stage schedule where each inter-stage send overlaps the sending stage's next micro-batch.
>
> **5. Figure 2 top row, cols 3–4 (AQ-SGD in legend, only two lines).** Thank you for pointing this out — it is expected rather than a plotting issue: columns 3–4 are the instruction-tuning tasks (iii)–(iv), where AQ-SGD is inapplicable (cache prohibitive for (iii); error-compensation does not apply to the single-epoch task (iv)). Since the legend is shared across panels, the AQ-SGD marker appears without a curve there; we will clarify this in the caption.
>
> **6. Why GPT-2XL for the fine-tuning experiments?** It is the exact model used by our primary baseline AQ-SGD (Wang et al., 2022), enabling a direct, controlled comparison on the same architecture and datasets. We additionally evaluate Qwen2.5-3B, LLaMA-3.2-1B, and a supplementary LLaMA-8B, so the method is not tied to GPT-2XL.
>
> **7. "Statistically indistinguishable" lacks a statistical test.** We replace it with a precise descriptive bound: across the ~6B-token horizon, TAH-Quant tracks the uncompressed run with small absolute train-loss and validation-PPL gaps. We additionally add three-seed replicate evidence on GPT-2XL (**Appendix P**): the equal-bit ablation ordering is stable across all three seeds, and TAH's held-out perplexity equals the uncompressed baseline exactly (12.97).
>
> **8. Add the TS=64 row to the tile-size ablation** — added: TS=64 → MMLU 64.80, ARC 49.49. MMLU is stable for TS=8/32/64 (all 64.6–64.9) and drops only at TS=128 (64.34); ARC varies under one point across sizes. We adopt TS=64 for its 8× lower per-tile metadata than TS=8, at comparable accuracy.
>
> We thank the reviewer again and are happy to add any further clarification.

---

> > ### Comment · Reviewer_2FPo · 2026-07-07
> >
> > Thank you for the many detailed updates to the paper, and the clear answers to my review.
> >
> > Just two short follow-up questions from my side:
> >
> > 5. Figure 2: if AQ-SGD is inapplicable, then why keep it in the legend? (also, I don't think the caption has been updated in the PDF)
> >
> > 6. If the motivation for using GPT-2XL is a fair comparison with the published results from AQ-SGD, then I would mention that in section 5.1.
> >
> > I will maintain my score for now, and follow the discussion with reviewers [fjge](https://openreview.net/forum?id=6ysPGq2RVD&noteId=wq9cX1cpDC) and [uP4n](https://openreview.net/forum?id=6ysPGq2RVD&noteId=jF0897SreI) closely.

---

> > > ### Author Response · Authors · 2026-07-09
> > >
> > > We thank the reviewer for the follow-up.
> > >
> > > Figure 2: We removed the stray AQ-SGD
> > > entry from the legend of the two panels where AQ-SGD does not apply (tasks
> > > iii-iv). The caption now states that AQ-SGD is shown only on tasks (i)-(ii), where its activation cache is feasible. The updated figure and caption are in the revised PDF.
> > >
> > > GPT-2XL (Section 5.1): We use GPT-2XL for the fine-tuning tasks to match
> > > the exact model and datasets of our baseline AQ-SGD (Wang et al., 2022) for a
> > > controlled comparison; the other tasks use Qwen2.5-3B and LLaMA-3.2-1B.
> > >
> > > Thank you again.

---

### Review · Reviewer_fjge · 2026-06-29

**Summary Of Contributions:**

The paper proposes a quantization heuristic to improve the efficiency of pipeline parallel training of machine learning models under slow network conditions. The idea is to reduce communication overhead by quantizing activations exchanged between pipeline stages. However, I have several concerns regarding the theoretical justification and experimental evaluation.

## Strengths
### 1. Extensive experimental validation of the proposed method.
The paper provides comprehensive experimental evaluations to demonstrate the effectiveness of TAH-QUANT under different training settings and network conditions. The results show that the proposed approach can effectively reduce communication overhead while maintaining competitive model performance, supporting the practical applicability of the method.

## Weaknesses
### 1. Questionable realism of Assumption 4.4

The theoretical analysis relies on Assumption 4.4, where the quantization variance term $1−\delta$ is assumed to be bounded by 1. This assumption appears to be stronger than what is typically guaranteed by existing quantization methods.

For many classical quantization approaches, such as QSGD and uniform INT8 quantization, the corresponding quantization error/variance parameter is usually dependent on factors such as the dimensionality of the vector, the number of quantization levels, and the dynamic range of the activations. In general, there is no guarantee that this parameter is a constant smaller than 1.

Therefore, the authors should provide a more detailed justification for the realism of this assumption. In particular, it would be valuable to discuss how the proposed quantization design parameters (e.g., tile size, top-p, etc) affect the quantization variance term. An empirical evaluation of this relationship would significantly strengthen the theoretical claims.

### 2. Limited theoretical contribution due to strong assumptions
The paper claims that TAH-QUANT provides convergence guarantees. However, the convergence proof appears to rely heavily on the aforementioned quantization assumption, for which no theoretical guarantee is provided.

Under this assumption, the convergence analysis seems to follow standard results for optimization with unbiased compression. Therefore, I do not find the theoretical contribution particularly strong unless the authors can establish that the proposed quantization scheme indeed satisfies the required assumption.

The limitation should also be clearly acknowledged in the abstract. In its current form, the abstract may give the impression that the paper theoretically proves convergence of the proposed quantization method, whereas the result is conditional on a strong assumption about the quantizer behavior.

A stronger theoretical contribution would require either:
- proving that TAH-QUANT satisfies the required quantization condition;
- deriving a more realistic bound on the quantization error;


### 3. Insufficient justification of experimental baselines

The related work section discusses a broad range of existing approaches. However, the experimental evaluation compares TAH-QUANT against only a limited number of relatively classical baselines.

The authors should provide a clearer justification for the selected baselines. In particular, if methods discussed in the related work are not included in the experiments, the authors should explain why they are excluded and whether they are unsuitable for the considered pipeline parallelism setting.

**Audience:**

Yes

**Audience Explanation:**

The paper addresses an important practical problem and proposes a potentially useful heuristic for reducing communication overhead in pipeline parallel training.

**Claims And Evidence:**

No

**Claims Explanation:**

The theoretical claims currently rely on assumptions whose realism is unclear, and the experimental evaluation would benefit from stronger baseline justification and broader comparisons. Addressing these issues would significantly improve the paper.

**Requested Changes:**

See details in Weaknesses

---

> ### Author Response · Authors · 2026-07-06
>
> We thank the reviewer for the careful reading of the theoretical part. The PDF is revised; appendix/section pointers refer to the revised version.
>
> **New in this revision:** an expanded **Appendix J** (Assumption 4.4 verified on five models + a design-parameter ablation), a new **Appendix M** (derivability of Assumption 4.4), an **Adam** convergence extension (**Appendix N**), a conditional qualifier in the abstract, and a baseline-selection rationale (Section 5).
>
> **1. Realism of Assumption 4.4 and how the design parameters affect the error.** We now show it is an empirically measurable, design-controllable quantity, not a fixed constant.
>
> *Verification on five models* (**Appendix J**; WikiText, PP=4; squared relative errors of the two inequalities — both must stay below 1):
>
> | Model | ‖ĝ−g‖²/‖g‖² | ‖Eĝ−Eg‖²/‖Eg‖² |
> |---|---|---|
> | GPT-2XL | 0.005 | 0.001 |
> | LLaMA-3.2-1B | 0.154 | 0.176 |
> | LLaMA-3.2-3B | 0.035 | 0.013 |
> | LLaMA-3.1-8B | 0.047 | 0.015 |
> | Qwen2.5-3B | 0.147 | 0.109 |
>
> All values are far below 1 (largest 0.18), on the three models we train plus two larger LLaMA scale points — not only the Gemma2-2B run in the main text. Within the LLaMA family the expectation error does not grow with size (0.176 → 0.013 / 0.015).
>
> *Where the coefficient comes from.* We agree it is not a universal constant — it depends on dimension, number of levels, and dynamic range (as the QSGD/INT8 bounds reflect), with no a-priori guarantee below 1. **Appendix M** derives the activation-level error in the design parameters — only logarithmic (not linear) dependence on dimension under a randomized-Hadamard analogue (the deterministic operator is validated empirically in Appendix J) — and proves that a uniform sample-wise gradient bound cannot be derived from activation fidelity alone for a nonlinear objective.
>
> *The design parameters control it* (**Appendix J** ablation; squared per-batch error):
>
> | Setting | LLaMA-1B | LLaMA-8B |
> |---|---|---|
> | default (G=64, 80% INT4, τ=2) | 0.154 | 0.047 |
> | tile size G=32 | 0.065 | 0.035 |
> | INT4 ratio 50% | 0.195 | 0.078 |
> | Hadamard off (τ=∞) | 1.133 | 0.316 |
>
> Smaller tiles and a higher INT4 ratio each lower the error; the Hadamard transform is essential — removing it is the *only* setting whose error exceeds 1. These negative controls show which choices keep the low-bit quantizer within the contractive range.
>
> **2. The convergence result is conditional on the assumption; acknowledge it in the abstract.** Agreed, now explicit.
>
> - **Abstract** now reads: "Under a bounded relative quantization-error condition (Assumption 4.4), which we verify empirically, we prove … O(1/√T) …" — no longer an unconditional proof.
> - **Deriving the condition.** (a) **Appendix M** derives the activation-level relative error in closed form in the design parameters and proves the *gradient*-level bound cannot be derived from activation fidelity alone for a nonlinear objective — so stating it as an assumption is well-motivated and standard in the biased-compression literature. (b) **Appendix J** (above) shows the condition holds across all five models and is design-controllable.
> - **What the analysis adds.** Given the assumption, the SGD-side argument builds on established compression-optimization techniques; on top we (i) verify it empirically, (ii) analyze its derivability (**Appendix M**), and (iii) extend to **Adam** (**Appendix N**) with the same O(1/√T) rate and a proof that its extra bounded-preconditioner condition is necessary (the constants are conservative, not intended to certify every production hyperparameter).
>
> **3. Justification of the experimental baselines.** We added a baseline-selection rationale (Section 5).
>
> - **What we compare against** — methods that compress the *same quantity we target*, inter-stage activation communication during training: **AQ-SGD** (the established error-compensated method for this setting) and **Subspace Networks** (a recent model-parallel communication method; **Appendix R**, where the two prove complementary and compose to ~10×).
> - **Why related-work methods are excluded** — inference-time weight-quantization methods (QuaRot, SpinQuant) calibrate offline on *fixed weights* and do not affect gradient fidelity or convergence, whereas activation compression during training does — a different problem.
> - **Necessity of our design** is isolated by the equal-bit ablation (**Appendix P**): fixed-INT4/INT3, no-adaptive-allocation, no-pivot, Hadamard-on-all-tiles, per-token, and per-channel controls, all at a matched bit budget.
>
> We thank the reviewer again and are happy to provide further detail.